# Reduced proteasome activity in the aging brain results in ribosome stoichiometry loss and aggregation

Erika Kelmer Sacramento[1,†] (ID), Joanna M Kirkpatrick[1,†,‡], Mariateresa Mazzetto[1,2,†], Mario Baumgart[1], Aleksandar Bartolome[1], Simone Di Sanzo[1], Cinzia Caterino[1,2], Michele Sanguanini[3] (ID), Nikoletta Papaevgeniou[4], Maria Lefaki[4], Dorothee Childs[5], Sara Bagnoli[2], Eva Terzibasi Tozzini[2], Domenico Di Fraia[1], Natalie Romanov[5,§], Peter H Sudmant[6] (ID), Wolfgang Huber[5], Niki Chondrogianni[4], Michele Vendruscolo[3] (ID), Alessandro Cellerino[1,2,*] (ID) & Alessandro Ori[1,**] (ID)

## Abstract

A progressive loss of protein homeostasis is characteristic of aging and a driver of neurodegeneration. To investigate this process quantitatively, we characterized proteome dynamics during brain aging in the short-lived vertebrate *Nothobranchius furzeri* combining transcriptomics and proteomics. We detected a progressive reduction in the correlation between protein and mRNA, mainly due to post-transcriptional mechanisms that account for over 40% of the age-regulated proteins. These changes cause a progressive loss of stoichiometry in several protein complexes, including ribosomes, which show impaired assembly/disassembly and are enriched in protein aggregates in old brains. Mechanistically, we show that reduction of proteasome activity is an early event during brain aging and is sufficient to induce proteomic signatures of aging and loss of stoichiometry *in vivo*. Using longitudinal transcriptomic data, we show that the magnitude of early life decline in proteasome levels is a major risk factor for mortality. Our work defines causative events in the aging process that can be targeted to prevent loss of protein homeostasis and delay the onset of age-related neurodegeneration.

**Keywords** aging; lifespan; proteome; stoichiometry; transcriptome
**Subject Categories** Proteomics; Translation & Protein Quality
**Mol Syst Biol. (2020) 16: e9596**

## Introduction

Although age is the primary risk factor for cognitive decline and dementia (Assoc, 2018), the associated age-dependent molecular changes are still not known in detail. Despite the presence of clear functional impairments (Buckner, 2004), physiological brain aging is characterized by limited loss of neurons (Schmitz & Hof, 2007) and specific morphological changes of synaptic contacts (Dickstein *et al*, 2013). Large collections of data for transcript dynamics in human and animal brains indicate that systematic, age-dependent changes in gene expression are also relatively minor (Cellerino & Ori, 2017), although some shared transcriptional signatures have been identified, including a chronic activation of cellular inflammatory response (Aramillo Irizar *et al*, 2018), reactive changes in glial cells (Clarke *et al*, 2018), and reduced expression of neuronal and synaptic genes (Lu *et al*, 2004; Somel *et al*, 2010).

Since the vast majority of human neurons are generated during fetal and perinatal life and neuronal turnover is limited in the postnatal human brain (Sorrells *et al*, 2018), neurons are particularly prone to accumulate misfolded proteins that are not properly processed by the cellular proteolytic mechanisms (proteasomal and autophagic pathways), thus forming aberrant deposits. Indeed, neurodegenerative diseases are characterized by the prominent presence of protein aggregates, in particular due to mutations that facilitate misfolding and aggregation, and impairment of cellular quality control systems (Soto & Pritzkow, 2018). Accumulation of protein aggregates occurs also during physiological aging, as demonstrated

1   Leibniz Institute on Aging–Fritz Lipmann Institute (FLI), Jena, Germany
2   Bio@SNS, Scuola Normale Superiore, Pisa, Italy
3   Centre for Misfolding Diseases, Department of Chemistry, University of Cambridge, Cambridge, UK
4   Institute of Chemical Biology, National Hellenic Research Foundation, Athens, Greece
5   European Molecular Biology Laboratory, Heidelberg, Germany
6   University of California Berkeley, Berkeley, CA, USA
    *Corresponding author. Tel: +39 0503 152756; E-mail: alessandro.cellerino@sns.it
    **Corresponding author. Tel: +49 3641 65 6808; E-mail: alessandro.ori@leibniz-fli.de
    †These authors contributed equally to this work
    ‡Present address: Proteomics Science Technology Platform, The Francis Crick Institute, London, UK
    §Present address: Max Planck Institute of Biophysics, Frankfurt am Main, Germany

by the presence of lipofuscin (Glees & Hasan, 1976) and ubiquitinated cellular inclusions (Zeier et al, 2011; Matsui et al, 2019). However, the exact composition of these spontaneous aggregates and the mechanisms triggering their formation during brain aging remain unknown.

Although age-dependent transcript changes in the brain have been studied extensively (Blalock et al, 2003; Lu et al, 2004; Loerch et al, 2008; Colantuoni et al, 2011; Wood et al, 2013), we are just beginning to understand the corresponding global regulation of the proteome during aging (Somel et al, 2010; Ori et al, 2015; Walther et al, 2015). Substantial post-transcriptional regulation takes place in the aging brain, with a sizeable proportion of proteins being up- or down-regulated in the absence of changes in the levels of the corresponding transcripts (Ori et al, 2015), resulting a progressive mRNA-protein decoupling (Janssens et al, 2015; Wei et al, 2015). Protein aggregation could play a role in generating an imbalance between protein and transcript levels, but these aspects have not yet been investigated systematically in vertebrate brains.

To address this challenge, we studied the annual killifish Nothobranchius furzeri, which is the shortest-lived vertebrate that can currently be bred in captivity. With a median lifespan of 3–7 months (Valdesalici & Cellerino, 2003; Terzibasi et al, 2008; Ripa et al, 2017; Hu & Brunet, 2018), it has emerged as a convenient model organism to investigate genetic and non-genetic interventions on aging (Harel et al, 2015; Cellerino et al, 2016; Kim et al, 2016; Platzer & Englert, 2016; Ripa et al, 2017), since it replicates many typical aspects of vertebrate brain aging at the levels of behavior (Valenzano et al, 2006a,b), neuroanatomy (Tozzini et al, 2012), and global gene expression (Baumgart et al, 2014; Aramillo Irizar et al, 2018). Age-dependent processes are enhanced in this species, thus facilitating the detection of differentially expressed genes as compared to other model organisms (Wood et al, 2013; Baumgart et al, 2014; Frahm et al, 2017). Importantly, an age-dependent formation of inclusion bodies containing α-synuclein and spontaneous degeneration of dopaminergic neurons has been recently described in killifish (Matsui et al,

2019). This phenotype closely mimics human pathologies and make killifish an extremely attractive vertebrate system to study age-related neurodegenerative disorders and therapeutic strategy against them.

In this work, we applied RNA-seq, mass spectrometry-based proteomics, and analysis of protein aggregates in killifish of different ages to delineate a timeline of molecular events responsible for loss of proteome homeostasis during brain aging. In particular, we set to identify the nature and biophysical properties of proteins that preferentially aggregate in old brains, to comprehensively investigate the loss of stoichiometry of protein complexes and their assembly state, and the role played by the proteasome as an early driver of protein homeostasis collapse using in vivo pharmacological experiments. Finally, we tested whether interindividual differences in proteasome decline influence mortality.

## Results

### Transcript and protein levels become progressively decoupled during brain aging

We initially analyzed whole brains from animals of three different age groups by liquid chromatography–tandem mass spectrometry using a label-free method. Based on previous phenotypic data, we chose to compare young, sexually mature fish (5 weeks post-hatching, wph), adult fish (12 wph) that do not show aging phenotypes (Terzibasi et al, 2008), and old fish (39 wph) that display neurodegeneration (Di Cicco et al, 2011; Tozzini et al, 2012) (Fig 1A and Dataset EV1). Principal component analysis separated samples according to the age groups (Fig 1B). In order to achieve higher proteome coverage, we split the age groups into two separate experiments based on tandem mass tag (TMT) multiplexing, where we compared adult vs young fish and old vs adult fish (Fig EV1A). This was necessary because of the limited number of channels available (10 per experiment) and to do not

---

**Figure 1. Transcript and protein levels become decoupled during *Nothobranchius furzeri* brain aging.**

A Survival curve of *N. furzeri* in the FLI facility. Recording of deaths starts at age of 5 wph, which corresponds to sexual maturity, and the colored dashed lines indicate the three age groups analyzed in this study (five animals/group), namely 5 weeks post-hatching (wph, young, sexual maturity), 12 wph (adult), and 39 wph (old, past median lifespan) of a wild-derived strain that exhibits a median lifespan of 7–8 months.

B Principal component analysis (PCA) of brain samples based on the abundance of all proteins identified by label-free mass spectrometry. The smaller dots represent individual samples and the larger dots the centroids of each age-matched group. Ellipses represent 95% confidence intervals. The percentage of variance explained by the first two PC axes is reported in the axis titles.

C Global protein–transcript correlation for each sample, grouped by age. RPKM and iBAQ values were used to estimate transcript and protein levels from matched RNA-seq and TMT-based proteomics data obtained from the same animal. An ANOVA test was performed to evaluate significance among the age groups (mean correlation at 5 wph: 0.48; at 12 wph: 0.43; and at 39 wph: 0.33; $P = 3.05e{-}07$, $n = 5$ per age group). In boxplots, the horizontal line represents the median, the bottom, and top of the box the $25^{th}$ and $75^{th}$ percentile, respectively, and the whiskers extend 1.5-fold the interquartile range.

D, E Scatter plot of $\log_2$ fold changes for genes differentially expressed both at transcript and protein levels (adj. $P < 0.05$). The color gradients indicate gene density in the regions where individual points overlap. Numbers of genes in each quadrant and the value of Pearson's coefficient of correlation, $r$, are reported for each graph. Solid lines represent a spline fit ($r = 0.505$ for genes significantly affected at both transcript and protein levels, $P < 2.2 \times 10^{-16}$, D; $r = 0.126$, $P = 0.007$, E).

F Mechanisms affecting protein abundance during aging. Barplots are based on all the proteins affected in either one of the age comparisons (adj. $P < 0.05$). Proteins were divided into following five groups: (i) proteins and transcripts with significant and consistent changes (dark brown), (ii) proteins with significant changes, and with consistent changes of the transcripts (light brown), (iii) proteins with no transcripts detected (dark gray), (iv) proteins with transcripts whose translation is potentially regulated by miRNAs (light gray), as assessed by the workflow displayed in Fig EV2D, and (v) all the remaining proteins that we classified as regulated by other post-transcriptional mechanisms (violet). pgs, protein groups.

G, H Barplots representing enriched KEGG pathways among genes that showed significant changes at both transcript and protein levels in aging. Genes were grouped according to the four possible patterns of transcript and protein regulation, as visualized by their positions in the four quadrants shown in (D) and (E), respectively. Only pathways significantly enriched (FDR < 0.05) are shown. The complete list of enriched pathways is reported in Dataset EV4.

Data information; Related to Figs EV1 and EV2, Table EV1, and Dataset EV1–EV4.

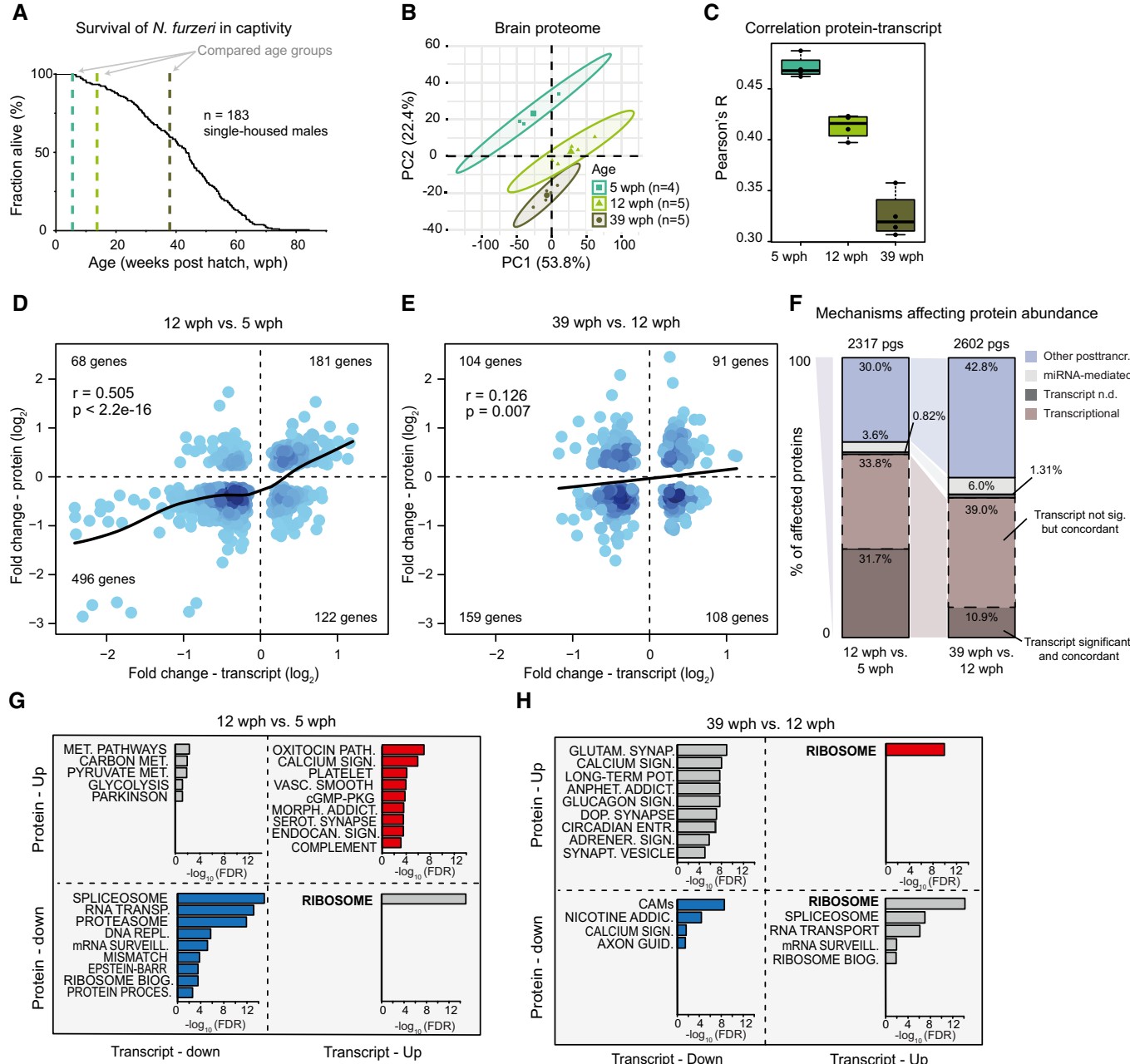

**Figure 1.**

reduce the number of animals analyzed per age group. A total of 8,885 protein groups were quantified with at least two proteotypic peptides, of which 7,200 were quantified in both experiments (Fig EV1B and Table EV1). Almost half of the quantified protein groups (4,179/8,885) was significantly affected by aging in at least one of the age comparisons (Dataset EV2). Functionally related proteins showed different patterns of abundance change between age groups, and pathways affected by aging in other species, including inflammation-related pathways (Aramillo Irizar *et al*, 2018), the complement, and coagulation cascade (Clarke *et al*, 2018), were affected in killifish already in the transition from young to adult (Fig EV1C and Dataset EV2).

Total RNA-seq after rRNA depletion and microRNA-seq were obtained from the same samples (Fig EV1D, Table EV1 and Dataset EV3). For each sample, absolute protein abundances estimated from peptide intensities (iBAQ values, (Schwanhäusser *et al*, 2011)) were correlated with the corresponding transcript levels obtained by RNA-seq (RPKM values), obtaining global protein–transcript correlation values for each sample separately. We observed a progressive age-dependent reduction of protein–transcript correlation values (Fig 1C), consistent with a decoupling between RNA transcripts and proteins during brain aging (Wei *et al*, 2015). Decoupling was observed also when analyzing an independent RNA-seq dataset from polyA$^+$ RNA for animals

of the same age groups (Baumgart *et al*, 2014) (Fig EV1E). Fold changes of genes differentially expressed in the two RNA-seq datasets were strongly correlated (Fig EV1F). For further analysis, we then focused on the dataset with higher sequencing depth and larger number of replicates, for which the absolute number of differentially expressed genes was higher (polyA$^+$ RNA dataset, Fig EV1G).

Direct comparison of protein and mRNA fold changes across age groups revealed discrepancies between RNA and protein regulation (Fig 1D and E and Dataset EV4). Protein and transcript changes were significantly correlated in the adult vs young fish comparison (Fig 1D), but the correlation was reduced in the old vs adult comparison (Fig 1E), further supporting a progressive decoupling between transcript and protein regulation. For validation, we analyzed proteins previously identified to be very long-lived in rodent brain (Toyama *et al*, 2013), including histones, collagens, and myelin proteins. For these proteins, we found that transcript, but not protein levels, were generally decreased in old fish, indicating that protein stability might contribute to the observed discrepancies between transcripts and proteins (Fig EV2A). In contrast, protein levels estimated by mass spectrometry and immunoreactivity for the glial fibrillary acidic protein (GFAP) were shown to increase significantly in the aging brain (Tozzini *et al*, 2012), while RNA levels remained unchanged (Fig EV2B). To exclude biases deriving from changes in cellular composition of the brain with aging, we analyzed the regulation of established cell markers (Sharma *et al*, 2015) in the age group comparisons. We found only minor changes that were consistent at the protein and transcript level (Fig EV2C), thus excluding that the observed decoupling between transcript and protein levels is due to changes in cellular composition.

To find out whether microRNAs could contribute to transcript-protein decoupling, we first analyzed miRNA expression levels across the same three age groups (Dataset EV3) and then mapped the targets of age-affected miRNAs to our proteome data (Fig EV2D). By considering potential regulation mediated by miRNAs, we defined a subset of proteins whose abundance is affected by aging via mechanisms independent of both transcript level and miRNA-mediated post-transcriptional regulation (Fig 1F). This subset accounted for 30% of the affected proteins in the adult vs young fish comparison and increased up to 43% in the old vs adult comparison.

To clarify whether the transcript-protein decoupling preferentially affects some specific pathways, we classified age-affected genes according to their respective transcript and protein fold changes, and performed pathway overrepresentation analysis (Dataset EV4). In the comparison adult vs young fish, pathways related to the complement coagulation cascade and synaptic function/plasticity were overrepresented in concordantly increased transcripts and proteins (Fig 1G), in agreement with the notion that synaptogenesis continues during this phase of residual brain growth (Tozzini *et al*, 2012). By contrast, genes coding for biosynthetic pathways such as RNA transport, splicing and surveillance of RNA, ribosome biogenesis, and protein processing in the endoplasmic reticulum (ER) were overrepresented in concordantly decreased proteins/transcripts (Fig 1G). These changes may be related to the reduction of adult neurogenesis (Tozzini *et al*, 2012) that accounts for a significant fraction of global transcription regulation occurring

during this period (Baumgart *et al*, 2014). The same biosynthetic pathways become discordant when old and adult animals are compared, with protein levels decreasing further with age, while transcript levels changed directionality and increased (Fig 1H, bottom right quadrant).

Taken together, our data indicate that post-transcriptional mechanisms regulating protein levels have an increasingly important role in modulating protein abundance with age; they are responsible for nearly half of the protein changes observed in old brains, and they can lead to the opposite regulation trends for proteins and mRNAs.

## Loss of ribosome stoichiometry in the aging brain

Among genes showing opposite transcript and protein changes already in the adult vs young fish comparison, we identified 13 genes encoding ribosomal proteins with transcript levels being significantly increased and protein abundances decreased (Figs 2A and EV3A). Fold changes of genes encoding for ribosomal proteins split into two groups in the old vs adult fish comparison (Fig 1H): While transcript levels continue to increase consistently, ribosomal proteins show either increased (13 proteins, e.g., RPS20, RPL8, and RPL21) or decreased (14 proteins, e.g., RPS6, RPLP2, and RPL22L1) abundance (Figs 2A and EV3A). A similar pattern was observed also for the mitochondrial ribosome (Fig EV3B). These findings indicate a loss of stoichiometry of ribosomal proteins (i.e., an imbalance in their relative levels) during aging, which is likely to impair ribosome assembly and to create a pool of orphan proteins at risk of aggregation. When mapped on the ribosome structure (Khatter *et al*, 2015), age-affected proteins form clusters of either consistently increased or decreased abundance (Fig 2B). Since transcript level changes are consistent, while ribosomal protein levels are not (Figs EV3A and B), the loss of ribosome stoichiometry must result from an alteration of post-transcriptional mechanisms mediating protein homeostasis. We obtained ribosome footprint data from young and old killifish brains and showed consistently increased levels of transcript encoding for ribosomal proteins to be associated with ribosomes in old brains, as previously shown in rats (Ori *et al*, 2015) (Fig EV3C). These data exclude changes in translation output as a cause for the observed loss of ribosome stoichiometry and point to other mechanisms such as protein degradation and aggregation.

In order to directly investigate the consequences of stoichiometry loss on the assembly state of ribosomes, we performed size-exclusion chromatography of brain lysates coupled to data independent acquisition (DIA) quantitative mass spectrometry on two pools each of young and old killifish (Fig 2C). Our analysis retrieved known protein complexes as distinct co-eluting peaks in both young and old brains (Fig 2D and E, Dataset EV5). Interestingly, we found protein components of the ribosome to co-elute at lower than expected molecular weight in old brain lysate. This effect was particularly pronounced for the large cytoplasmic ribosome and the mitochondrial ribosome (Fig 2F). Other complexes were not affected and eluted at the same retention time in both young and old lysates (Fig 2E), pointing to a specific effect on ribosomes. Taken together, these data indicate that age-dependent loss of stoichiometry of ribosomes might derive from altered assembly/disassembly in old brains.

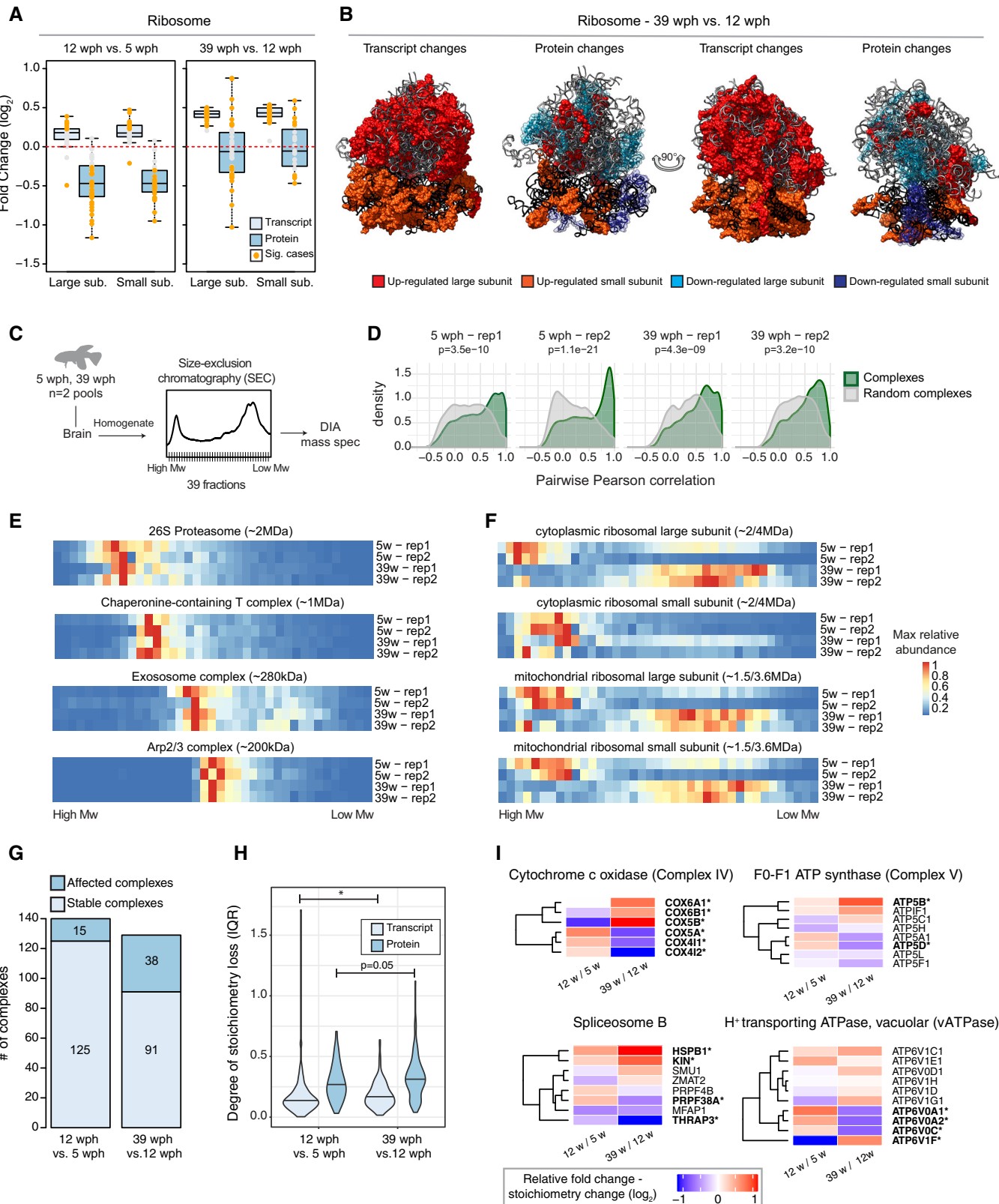

**Figure 2.**

**Figure 2.  Loss of stoichiometry and disassembly of ribosomes in old killifish brain.**

A   Abundance changes of ribosomal proteins and their transcripts during aging. Cytoplasmic ribosomal proteins of large and small subunits are displayed separately, and changes are shown for both the age comparisons as boxplots. Transcripts are displayed as light blue and proteins as dark blue boxes. Changes of individual proteins are displayed as dots; orange dots identify significant cases (adj. $P < 0.05$, $n = 4$ per age group for proteome and $n = 5$ per age group for transcriptome). In boxplots, the horizontal line represents the median, the bottom and top of the box the 25th and 75th percentile, respectively, and the whiskers extend 1.5-fold the interquartile range.

B   Visualization of age-related changes of proteins and transcripts projected on the 80S ribosome complex structure. Ribosomal RNAs are depicted in ribbon form: 28S rRNA, 5S rRNA, and 5.8S rRNA of large subunit are depicted in light gray, and 18S rRNA of small subunit is depicted in black. Ribosomal proteins are depicted as molecular surfaces and shown only if significant changes in the level of corresponding mRNA or protein were detected. Affected proteins of large and small subunits are visualized in two different shades of red (up-regulated), or blue (down-regulated). For clarity, down-regulated components are displayed as transparent molecular surfaces. Visualization was performed with USCF Chimera (version 1.12), according to Protein Data Bank archive— human 80S ribosome 3D model: 4UG0.

C   Brains from young (5 wph) and old (39 wph) were homogenized and clarified lysates separated by size-exclusion chromatography (SEC). For each age group, two pools of brains were processed separately. For each experiment (four in total), 39 fractions were collected along the chromatogram, digested into peptides, and analyzed by data independent acquisition (DIA) quantitative mass spectrometry.

D   Co-elution of members of protein complexes in SEC. For each experiment, the distribution of pairwise correlations between members of the same protein complex was analyzed (green). As expected, members of protein complexes tend to co-elute in all SEC experiments, as indicated by positive correlation values. A set of randomly defined protein complexes was used as control (gray). For all the experiments, the correlations of real complexes are significantly higher than random ones, Wilcoxon rank-sum test.

E, F  Co-elution profiles for selected protein complexes. For each complex, the median abundance of all the quantified subunits was used to generate the complex profile across fractions (Dataset EV5). All the complex profiles are scaled to the max value (set to 1) to make profiles comparable across experiments. The estimated molecular weight of the displayed complexes is indicated in brackets.

G   Statistics of protein complexes undergoing stoichiometry changes with aging. Only protein complexes that had at least five members quantified were considered for each comparison. Complexes were considered affected if at least two members showed significant stoichiometry change (adj. $P < 0.05$ and absolute $\log_2$ fold change > 0.5). The complete list of stoichiometry changes is available in Dataset EV6.

H   Violin plots depicting interquartile ranges (IQRs) of individual members of protein complexes during aging. The IQR for each protein complex considered in G was calculated using transcript (total RNA dataset, light blue) or protein (dark blue) $\log_2$ fold changes between two age groups ($n = 4$ per age group for proteome and $n = 5$ per age group for transcriptome). *$P < 0.05$, Wilcoxon rank-sum test. The central line of the violin plots indicates the median value.

I   Heatmap showing relative protein fold changes for members of selected complexes affected by aging. Names of significantly affected members in the 39 vs 12 wph comparison (adj. $P < 0.05$ and absolute $\log_2$ fold change > 0.5) are highlighted in bold with a star.

Data information: Related to Fig EV3 and Dataset EV5 and EV6.

## Widespread stoichiometric imbalance in protein complexes during aging

We next asked whether the age-related loss of stoichiometry described above occurs more widely in the proteome. We thus analyzed all annotated protein complexes in the two age group comparisons (Ori *et al*, 2013, 2016). We found that the number of complexes undergoing stoichiometry changes increases from 11% (16 out of 140) between 5 and 12 wph to 30% (39 out of 129) between 12 and 39 wph (Fig 2G and Dataset EV6). Consistently, the number of affected complex members increases almost twofold in the old vs adult comparison (from 6 to 13%; Fig EV3D). Loss of stoichiometry was confirmed by an alternative metric, namely an increase in the interquartile range (IQR) of fold changes of protein complex members (Janssens *et al*, 2015) in the old vs adult fish comparison (Fig 2H). In order to exclude potential batch effects, we repeated the analysis on a subset of three animals for each age group that were analyzed in the same TMT10plex experiment and confirmed an age-dependent increase of IQR between protein complex members (Fig EV3E).

When individual complexes were ranked according to the difference of IQR between the two age comparisons, the majority of the complexes showed an increase in IQR (76 out of 124, 61%, Fig EV3F and Dataset EV6). The most affected complexes included Complex IV and Complex V, but not Complex I and Complex III, of the mitochondrial respiratory chain, the cytoplasmic ribosome, the 26S proteasome, the B complex of the spliceosome, and the lysosomal V-type ATPase (Fig 2I). These complexes take part in biological processes known to be causative in aging (Dillin *et al*, 2002; Lee *et al*, 2003; Chondrogianni *et al*, 2014; Carmona-Gutierrez *et al*, 2016; Steffen & Dillin, 2016; Heintz *et al*, 2017), and the regulation

of transcripts coding for them is correlated with individual lifespan in a longitudinal RNA-seq study in *N. furzeri* (Baumgart *et al*, 2016). Many of these complexes are affected by stoichiometry changes already at the adult stage (Dataset EV6), identifying these alterations as early events during aging progression.

## Age-dependent protein aggregates are enriched for ribosomal proteins

Loss of stoichiometry and altered assembly of protein complexes can create a pool of orphan proteins at risk of aggregation. Since protein aggregates are known to be SDS-insoluble (Reis-Rodrigues *et al*, 2012), we compared SDS-insoluble fractions from brain homogenates of young and old animals (Figs 3A and EV4A). We used mice for this analysis because of the larger brain size that allows retrieval of sufficient amount of aggregates (that constitute only about 0.5% of total proteins in old brains) for proteomic analysis. As expected, the yield of SDS-insoluble protein aggregates was significantly higher from old animals, confirming that aging is associated with enhanced protein aggregation (Figs 3B and EV4B). We then analyzed the aggregate composition by quantitative mass spectrometry to identify proteins enriched in these aggregates as compared to the starting total brain homogenates (Dataset EV7). Enriched proteins showed a predicted higher molecular chaperone requirement for folding and were richer in intrinsically disordered regions (Fig 3C). Among the enriched proteins, we found collagens (Col1a1, Col1a2, and Col4a2), which are well-known to undergo age-dependent crosslinking (Viidik, 1979), and ferritins (Fth1 and Ftl1), whose aggregation is linked to the age-dependent brain accumulation of intracellular iron (Ripa *et al*, 2017) (Fig 3D). Interestingly, protein aggregates were also

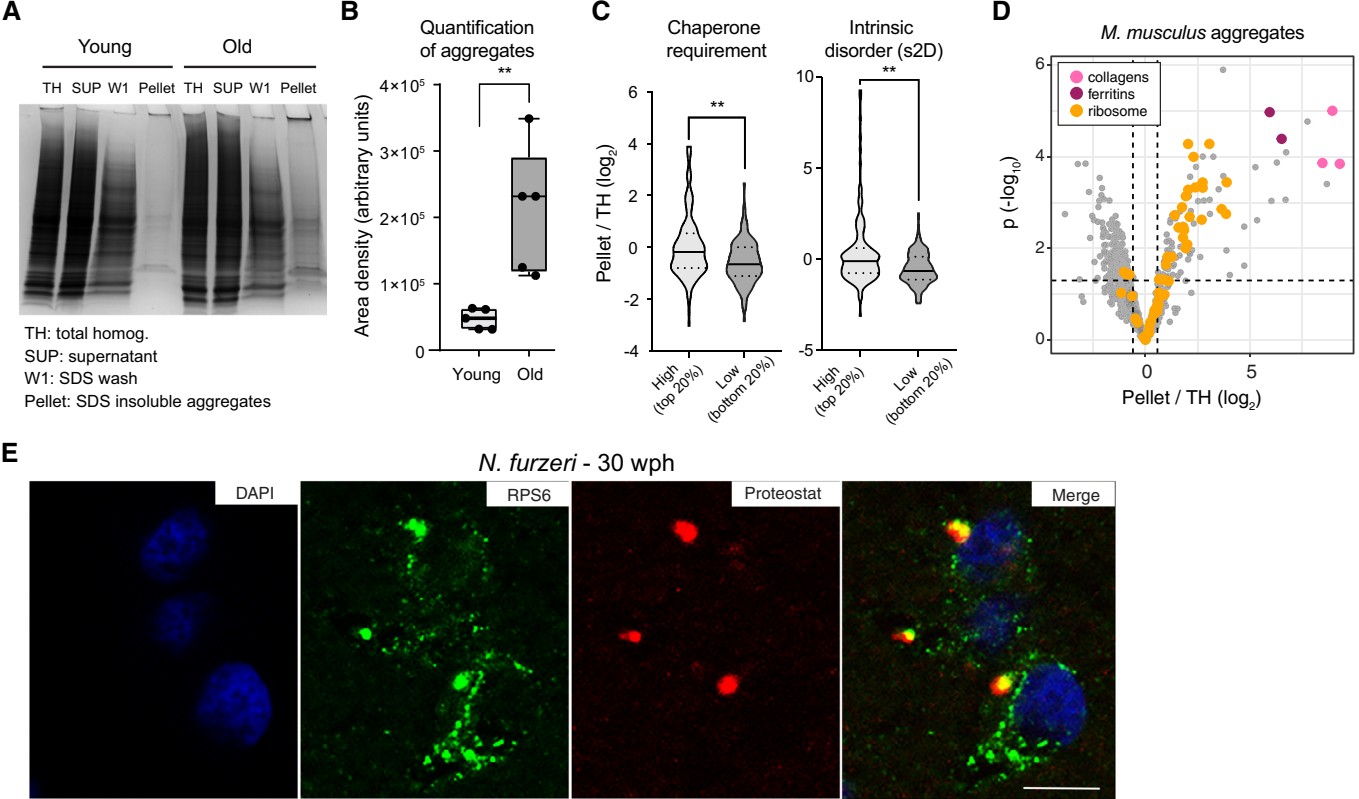

**Figure 3. Aggregation of ribosomal proteins during brain aging.**

A   Representative Coomassie-stained SDS–PAGE gel showing the isolation of SDS-insoluble aggregates from mouse brain lysates. SUP = supernatant, TH = total homogenate, W1 = SDS-soluble fraction, Pellet = formic acid soluble fraction (see Fig EV4A).

B   Quantification of the yield of SDS-insoluble aggregates from young and old brain lysates was based on densitometry analysis of Coomassie-stained gel bands obtained from different animals, *n* = 5 per age group (Fig EV4B); **$P$ < 0.01, unpaired *t*-test. In boxplots, the horizontal line represents the median, the bottom, and top of the box the 25th and 75th percentile, respectively, and the whiskers extend 1.5-fold the interquartile range.

C   Proteins enriched in aggregates show a predicted higher molecular chaperone requirement for folding (top vs bottom 20% $P$ = 0.0055, Kolmogorov–Smirnov test) and are richer in intrinsically disordered regions (s2D-derived scores, top vs bottom 20% $P$ = 0.0019, Kolmogorov–Smirnov test). Violin plots: The solid line shows the median and the dotted lines the interquartile ranges. The same result was obtained with cleverSuite-derived scores (top vs bottom 20% $P$ = 0.0201, Kolmogorov–Smirnov test; Fig EV4C). **$P$ < 0.01, Kolmogorov–Smirnov test.

D   Volcano plot based on protein quantification by label-free mass spectrometry depicting the enrichment of specific proteins in protein aggregates. The *x*-axis indicates the log₂ ratio between protein abundance in aggregates (Pellet) and starting total homogenate (TH). The horizontal dashed line indicates a $P$ value cut-off of 0.05 and vertical lines a log₂ fold change cut-off of ± 0.5. Selected proteins are highlighted as colored dots as indicated in the figure legend. Protein quantification was based on samples obtained from three independent isolations.

E   Double labeling of telencephalic sections of *Nothobranchius furzeri* with anti-RPS6 (green) as ribosomal marker and Proteostat as a marker for aggregated proteins (red). Nuclear counterstaining was performed with DAPI (blue). Scale bar = 10 μm.

Data information: Related to Fig EV4 and Dataset EV7.

enriched for ribosomal proteins ($P$ = 4.4e−05, Fisher test, Figs 3D and EV4D) that are both characterized by protein transcript decoupling (Fig 1H) and loss of complex stoichiometry (Fig 2A and B) during aging. Other protein complexes that displayed loss of stoichiometry (i.e., Complex V and vATPase) did not show significant enrichment in aggregates, indicating that stoichiometry imbalance does not always correlate with protein aggregation (Fig EV4D).

To confirm the aggregation of ribosomal proteins in killifish, we performed staining of young (7–10 wph) and old (27–30 wph) brain slices using Proteostat, an amyloid-specific dye (Shen *et al*, 2011). As expected, we detected lysosomal aggregates in old, but not in young brains (Fig EV4E, F, and H). These aggregates appeared to contain the ribosomal protein RPS6 (Figs 3E and EV4G), which was found to be significantly enriched in aggregates in mice (Dataset

EV7). In addition, we performed mass spectrometry on aggregates from killifish. Although the limited amount of material precluded a quantitative analysis as performed in mouse, we were able to confidently identify several ribosomal proteins also in killifish brain aggregates (Fig EV4I and Dataset EV7). Taken together, these data demonstrate that aggregation of ribosomal proteins is a conserved trait of brain aging in fish and mice.

## Acute partial reduction of proteasome activity is sufficient to induce loss of protein stoichiometry *in vivo*

Protein stoichiometry loss could be due to decreased proteolysis rates. We focused on the proteasome, which is one of the main degradation machineries and itself undergoes stoichiometry loss

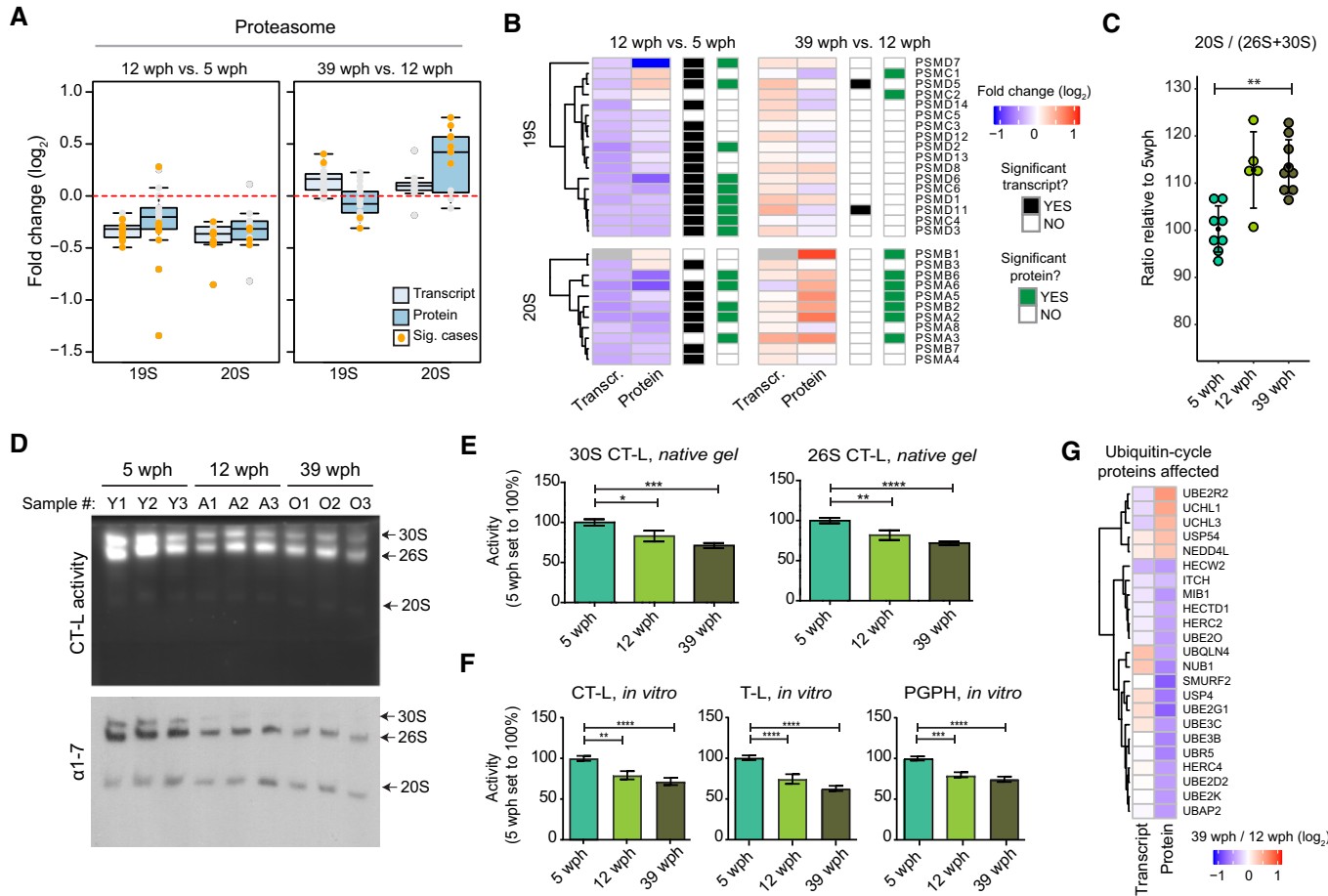

**Figure 4. Reduced proteasome activity and assembly in old brains.**

A  Abundance changes of proteasome proteins and their transcripts during aging. Members of the 19S and 20S complex are displayed separately, and changes are shown for both the age comparisons as boxplots. Transcripts are displayed as light blue and proteins as dark blue boxes. Changes of individual proteins are displayed as dots and orange dots represent significant cases (adj. $P < 0.05$, $n = 4$ per age group for proteome and $n = 5$ per age group for transcriptome). In boxplots, the horizontal line represents the median, the bottom, and top of the box the 25th and 75th percentile, respectively, and the whiskers extend 1.5-fold the interquartile range.

B  Heatmap showing transcript and protein fold changes for members of the 26 proteasome (19S and 12S complexes). Genes are annotated according to significance of their changes at the level of transcript (adj. $P < 0.05$: black, adj. $P > 0.05$: white) or protein (adj. $P < 0.05$: green, adj. $P > 0.05$: white).

C  The between 20S and 26S + 30S proteasome abundance assessed by immunoblot on native gels (Fig EV5A). $**P < 0.001$, Wilcoxon rank-sum test. The bars indicate mean ± SD.

D  In-gel proteasome assay following native gel electrophoresis (top) and immunoblotting of proteasome complexes (30S, 26S and 20S; bottom) in young (5 wph), adult (12 wph), and old (39 wph) killifish brains. For additional samples and low exposure pictures, see Fig EV5B and C.

E  Barplots depicting the quantification of chymotrypsin-like (CT-L) activity from native gels calculated for doubly capped (30S) or singly capped (26S) proteasomes. $n \geq 5$ per sample group; error bars indicate standard error of the mean. $*P < 0.05$, $**P < 0.005$, $***P = 0.0001$, $****P < 0.001$, one-way ANOVA, Holm–Sidak's multiple comparison test. For each sample group, the mean value of activity in young samples (5 wph) was set to 100%.

F  Percentage (%) of chymotrypsin-like (CT-L), trypsin-like (T-L), and peptidylglutamyl peptide hydrolyzing or caspase-like (PGPH) proteasome activities in brain extracts of killifish of different ages. $n \geq 6$ per sample group; error bars indicate standard error of the mean. $**P < 0.005$, $***P = 0.0001$, $****P < 0.001$, one-way ANOVA, Holm–Sidak's multiple comparison test. For each sample group, the mean value of each activity in young samples (5 wph) was set to 100%.

G  Age-related changes of proteins involved in the ubiquitin cycle. All the displayed proteins showed significant protein level changes in the 39 vs 12 wph comparison (adj. $P < 0.05$).

Data information: Related to Fig EV5.

upon aging in killifish (Fig EV3F). First, we observed a decrease in the levels of both proteasome proteins and their transcripts in adult killifish, which was followed by a stoichiometry imbalance that manifested in old fish exclusively at the protein level. In particular, we observed an imbalance between proteins belonging to the 19S and the 20S complexes, with the latter being exclusively up-regulated in old fish (Fig 4A and 4B). We confirmed an age-dependent

increase of 20S relatively to single- (26S) and double-(30S) capped proteasomes using immunoblots based on native gel electrophoresis from an independent cohort of samples (Fig 4C and EV5A).

To further investigate the proteasome functional status in killifish brains of different age groups, we performed native gel electrophoresis of proteasomes accompanied by in-gel proteasome activity assays (Chondrogianni *et al*, 2015). A significant decrease in the

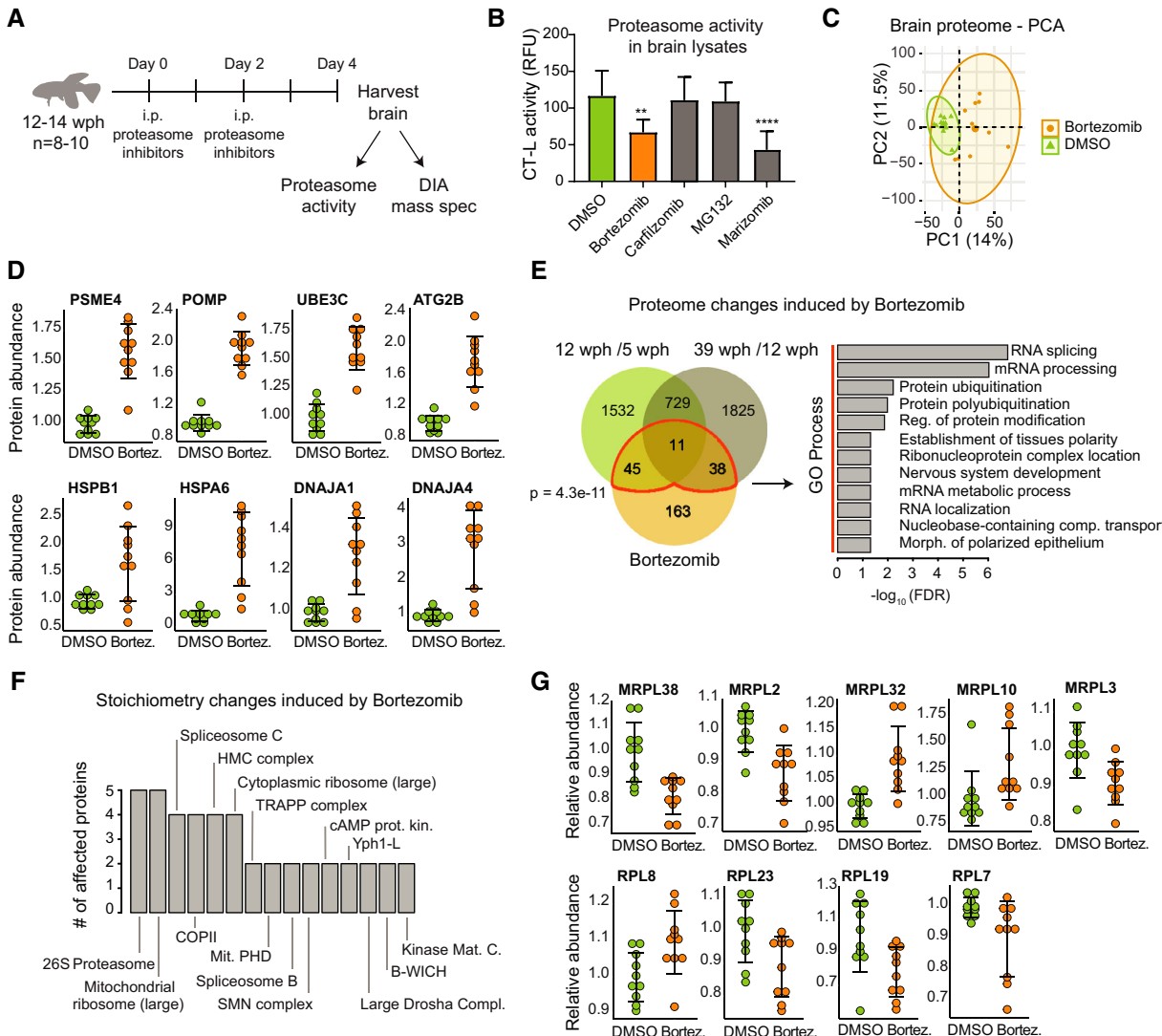

**Figure 5. Partial pharmacological inhibition of proteasome activity in adult killifish brain affects aging protein networks and induces stoichiometry changes in a subset of protein complexes.**

A Study design. Adult (12–14 wph) killifish were treated with different proteasome inhibitors or vehicle control for 4 days. Proteasome activity and proteome changes were analyzed in brains from treated and control fish.

B Chymotrypsin-like (CT-L) proteasome activity in brain extracts of killifish treated with different proteasome inhibitors. The activity was measured at day 4 after the beginning of treatment. $n = 8$ per sample group; error bars indicate standard deviation. **$P < 0.01$, ****$P < 0.0001$, one-way ANOVA followed by Holm–Sidak's multiple comparisons test.

C Principal component analysis (PCA) of brain samples based on proteome profiles obtained by data independent acquisition (DIA) quantification treated with Bortezomib or vehicle control (DMSO). $n = 10$ per sample group. The smaller dots represent individual samples and the larger dots the centroids of each age-matched group. Ellipses represent 95% confidence intervals. The percentage of variance explained by the first two PC axes is reported in the axis titles.

D Proteasome inhibition induces proteasome activators (PSME4), assembly factors (POMP) ubiquitin ligases (UBE3C), mediators of autophagosome formation (ATG2B), heat shock proteins, and chaperones in killifish brain. Protein abundances were quantified by DIA mass spectrometry, and they are shown relative to the mean value of vehicle control samples (DMSO) set to 1, $n = 10$ per sample group. Adj. $P < 0.05$ for all the displayed proteins. Error bars indicate mean ± SD.

E Overlap between proteins affected by Bortezomib treatment and aging in killifish brain. For all comparisons, only significantly affected proteins adj. $P < 0.05$ were considered. A significant overlap between Bortezomib and aging-affected proteins is detected (Fisher's test, as indicated in the figure panel). Significantly enriched GO biological process terms in the subset of overlapping proteins are indicated (FDR < 0.05).

F Bortezomib treatment affects the stoichiometry of a subset of protein complexes. The number of affected proteins (adj. $P < 0.25$) for each protein complex is indicated. Only protein complexes that had at least two members affected are shown.

G Members of the mitochondrial and cytoplasmic ribosomes affected by Bortezomib treatment. Relative protein abundances (normalized to the mean of the protein complex to which they belong) are shown. The mean value of vehicle control samples (DMSO) was set to 1. $n = 10$ per sample group. Adj. $P < 0.25$ for all the displayed proteins. Error bars indicate mean ± SD.

Data information: Related to Dataset EV8.

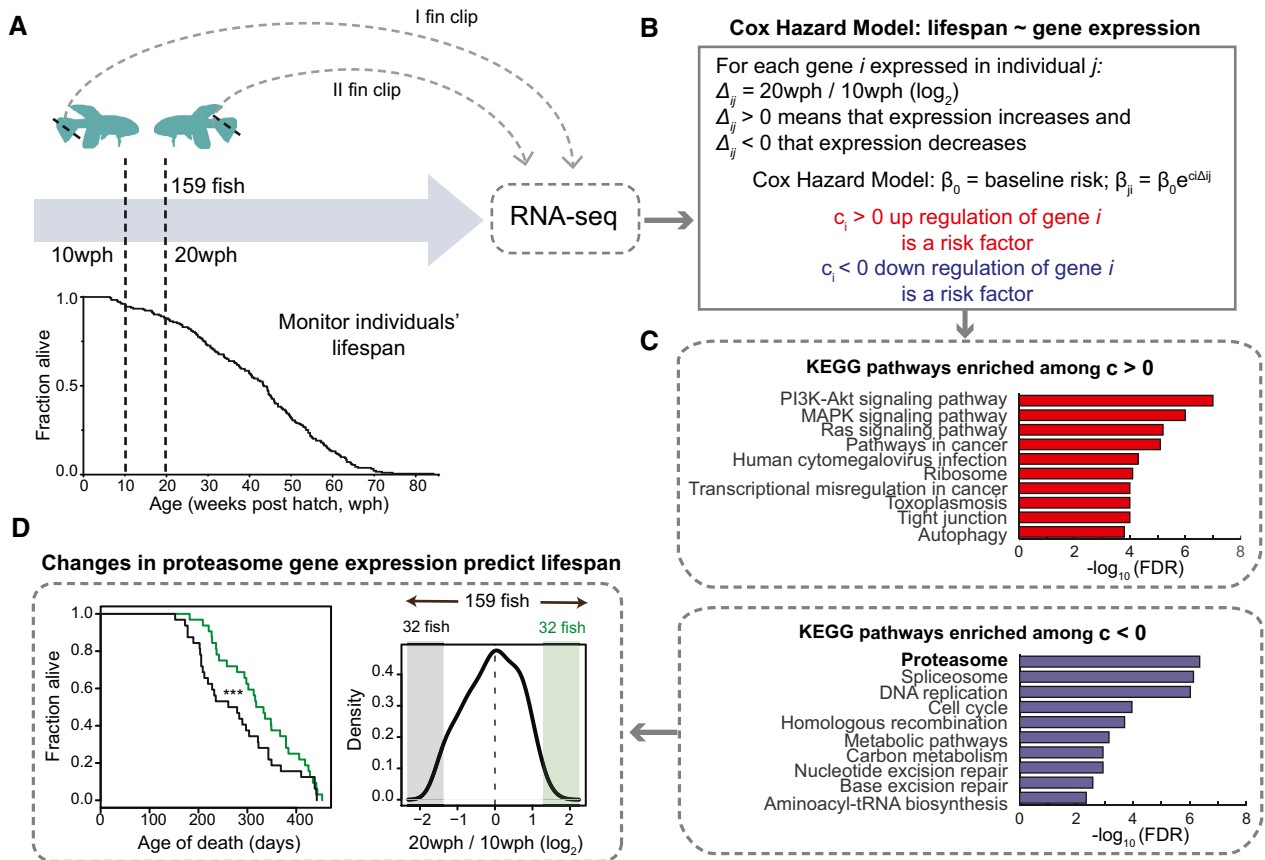

**Figure 6.** Longitudinal study in 159 killifish identifies early in life decrease of proteasome transcripts as a major risk factor for reduced lifespan.

A  Study design. Two fin clips were taken at 10 and 20 weeks post-hatching (wph) from 159 killifish and individual lifespans monitored. RNA sequencing was employed to compare transcriptome changes between 10 and 20 wph for each individual fish.

B  A Cox–Hazard model was used to correlate lifespan to gene expression changes. Two groups of genes were identified: (i) genes whose increased expression between 20 and 10 wph is a risk factors (i.e., associated to increased mortality risk; red) and (ii) genes whose decreased expression is a risk factors (blue).

C  KEGG pathways enriched among genes whose regulation is associated to mortality. Only pathways with FDR < 0.05 are shown.

D  Distribution of change in expression for proteasome transcripts across the entire cohort of 159 fish. Lifespan was compared among fish that showed extreme changes in proteasome levels between 10 and 20 wph (32 fish showing the most pronounced decreases, shown in black vs 32 fish showing the most pronounced increases). ***$P$ < 0.001 log-rank test.

Data information: Related to Dataset EV9.

levels of both 30S and 26S proteasomes was revealed already in adult animals (Figs 4D and EV5B and C). This decrease was accompanied by a significant reduction of all three proteasome activities in adult and old samples as compared to young samples (Fig 4E and F). Additionally, we detected down-regulation of enzymes involved in the ubiquitin cycle (ubiquitin-conjugating enzymes and ubiquitin ligases) in the old killifish brain (Fig 4G). These include the ubiquitin ligase UBE2O that has been shown to mediate recognition, ubiquitination, and targeting for proteasomal degradation of mislocalized ribosomal proteins (Yanagitani et al, 2017).

Next, we investigate whether an acute partial reduction of proteasome activity is sufficient to induce age-related phenotypes in killifish brain. Thus, we treated adult (12 wph) killifish with Bortezomib, a reversible proteasome inhibitor, for 4 days and achieved ~ 50% inhibition of proteasome activity in brain, mimicking the activity observed in old fish (Fig 5A and B). Quantitative mass spectrometry revealed distinct proteome changes induced by partial proteasome inhibition in brain (Fig 5C and Dataset EV8). These changes included a

compensatory up-regulation of proteasome activators (e.g., PSME4), ubiquitin ligases (e.g., UBE3C), autophagy-related proteins (e.g., ATG2B), and heat shock proteins (e.g., HSPB1; Fig 5D). Interestingly, this proteomic signature mimicked an aging phenotype, as indicated by a significant overlap with proteins whose abundance changes with age (Fig 5E). Using the same approach applied for aging data, we were able to detect protein complexes whose stoichiometry was affected by acute proteasome inhibition. These include the large subunits of both cytosolic and mitochondrial ribosomes (Fig 5F and G, and Dataset EV8). Taken together, these data indicate that decreased proteasome activity is an early event during brain aging that is sufficient to induce loss of stoichiometry of ribosomes in the brain in vivo.

### Early in life decrease of proteasome is a major risk factor for early death

To assess whether decline of proteasome levels is relevant for lifespan determination, we analyzed a longitudinal dataset of RNA-seq

comprising 159 killifish where transcripts from fin biopsies were quantified at 10 and 20 wph. Age-dependent variations in gene expression were related to the lifespan of individual fish (Fig 6A). By implementing a Cox–Hazard model, we identified genes for which the amplitude of age-dependent regulation significantly correlates with mortality risk (Fig 6B and Dataset EV9). We found the proteasome to be the most enriched category among genes predictive of lifespan (Fig 6C). In particular, a decrease of expression of proteasomal transcript between 10 and 20 wph increases mortality risk and is, thereby, associated with shorter lifespan. Accordingly, when we classified the 159 fish on the basis of changes in proteasome expression, we found that the lifespan of individuals showing the largest age-dependent down-regulation of transcripts coding for proteasomal proteins was significantly shorter than the lifespan of individuals showing the largest up-regulation (Fig 6D). These data support the finding obtained in the brain that the decrease of proteasome levels is an early event during aging and demonstrate that the rate of proteasome down-regulation in early adult life is predictive of lifespan in killifish.

# Discussion

### A molecular timeline for aging

Our results delineate a timeline of events associated with loss of protein quality control during aging. An early event, detectable already in adult fish, is a decreased proteolytic activity of the proteasome, which is driven by a down-regulation of transcripts coding for components of the 19S and 20S complexes in adult fish brain (Fig 4A and B). The amplitude of this down-regulation correlated with individual lifespan and could represent an early driver of aging (Fig 6D). The reduction of proteasome activity precedes chronologically the decoupling of transcript/protein levels, suggesting a causative role in this aspect of the aging process. Decreased proteasome activity can lead to the accumulation of proteins that are synthesized in excess relative to their binding partners, thus causing a stoichiometric imbalance of protein complexes (McShane *et al*, 2016). For instance, deletion of the ubiquitin ligase Tom1 (yeast homologue of Huwe1), which is responsible for the labeling for degradation of overproduced ribosomal proteins, leads to accumulation of multiple ribosomal proteins in detergent-insoluble aggregates in yeast (Sung *et al*, 2016). Accumulation of ribosomes in detergent-insoluble aggregates (David *et al*, 2010) and a loss of stoichiometry in the proteasome (Walther *et al*, 2015) were previously reported to occur during aging in *Caenorhabditis elegans*. Our work demonstrates the conservation of these mechanisms in the vertebrate brain, by showing alteration of stoichiometry in several large protein complexes (Fig 2G–I) and aggregation of ribosomes in old brain (Fig 3D and E). Specifically, we establish a mechanistic link between the partial reduction of proteasome activity observed in adult brains and the loss of stoichiometry of protein complexes (Fig 5F and G).

Later in life, the stoichiometry imbalance in protein complexes contributes to exacerbate the loss of protein homeostasis. Proteasome activity is further reduced in the old brain, correlating with an increased imbalance between the 19S and 20S complexes over time (Fig 4B and C). In addition, altered stoichiometry of the ribosome

can underlie both the reduction and the qualitative changes of protein synthesis in brain aging (Schimanski & Barnes, 2010; Ori *et al*, 2015; Sudmant *et al*, 2018). An alteration of the stoichiometry between membrane-bound and cytosolic components of the lysosomal v-type ATPase (Fig 2I) might influence the acidification of lysosomes and the activation of mTORC1 (Zoncu *et al*, 2011), thus hampering the clearance of protein aggregates. These aggregates in turn may further impair the proteasome activity (Grune *et al*, 2004), thus creating a negative feedback loop. The interconnectivity between proteasome and lysosome/autophagy system is further highlighted by the fact that partial inhibition of the proteasome in the adult brain induces mediators of autophagosome formation (e.g., ATG2B) and autophagy receptors (e.g., SQTSM/p62), likely as a compensatory mechanism to ensure removal of non-degraded proteins at risk of aggregation. It is tempting to speculate that a progressive decline of proteasome activity might be the trigger for the impairment of lysosomal function that characterize aging and late-onset neurodegenerative disorders (Wallings *et al*, 2019). The combination of reduced proteasome activity and impaired lysosome/autophagy would make old brains more vulnerable to the accumulation of protein aggregates, neuronal loss, and, consequently, favor the onset of neurodegenerative disorders.

Other key pathways implicated in aging are affected by loss of stoichiometry. In particular, alterations of respiratory chain complexes (particularly Complex IV and V) might contribute to their decreased activity and increased ROS production in old brain (Stefanatos & Sanz, 2018), and changes in multiple spliceosome complexes might underlie previously observed qualitative changes of splicing (Mazin *et al*, 2013; Ori *et al*, 2015). More detailed mechanistic studies are needed to demonstrate whether the alterations that we describe contribute to functional impairment of these protein complexes during aging or, rather, they represent adaptive responses to the aging process itself.

It remains to be determined which mechanisms promote the early decrease of proteasome activity in adult fish. Our data point to multiple processes being involved, including in particular: (i) decreased levels of rate-limiting proteasome members for the production of 20S assembled/functional proteasomes (e.g., PSMB5 or PSMB6, (Chondrogianni *et al*, 2015)), which we found to be significantly decreased already in the adult fish (Fig 4B); and (ii) changes in abundance of proteasome proteins that are important for the assembly and activity of the 19S proteasome complex, such as PSMD5. PSMD5 has been also shown to inhibit the assembly and activity of the 26S proteasome, and this proteasome member has been shown to be induced by inflammation (Shim *et al*, 2012). Accordingly, we detected also in killifish an activation of inflammation-related pathways (Fig EV1C) and, importantly, we identified PSMD5 as one of the few proteasome members to be up-regulated in the adult fish (Fig 4B). Finally, PSMD11 (known as RPN-6 in *C. elegans*), which has been shown to be responsible for increased proteasome assembly and activity in human embryonic stem cells and in *C. elegans* (Vilchez *et al*, 2012), was significantly down-regulated already in adult fish (Fig 4B). Identifying ways of counteracting these mechanisms might provide new avenues to delay organ dysfunction in aging and to increase lifespan. In this context, multiple studies have reported that transgenic animals (from various species) engineered to have enhanced proteasome activity show increased health- and lifespan (Vilchez *et al*, 2012; Chondrogianni

*et al*, 2015; Augustin *et al*, 2018) and that proteasome activity is preserved in cells from centenarians (Chondrogianni *et al*, 2000). Correspondingly, we have shown that expression level of proteasome genes predicts life expectancy in killifish (Fig 6C and D).

In conclusion, our work identifies the maintenance of proteasome activity upon aging as being critical to ensure the correct stoichiometry of protein complexes involved in key biological functions such as protein synthesis, degradation, and energy production.

# Materials and Methods

## Reagents and Tools table

| Reagent/Resource | Reference or source | Identifier or catalog number |
|---|---|---|
| **Experimental models** | | |
| C57BL/6JRj, *Mus musculus* | Janvier Labs | |
| *Nothobranchius furzeri* | https://www.leibniz-fli.de/research/cooperations/networking-projects/notho-project/ | |
| **Antibodies** | | |
| S6 Ribosomal Protein (RPS6) 1:100 | Cell Signaling | 2317 |
| Anti-LAMP1 1:500 | AbCam | Ab24170 |
| Anti-S100 1:400 | Dako | Z0311 |
| Anti-Glial Fibrillary Acidic Protein (GFAP) 1:800 | Dako | M0761 |
| Proteasome 20S α1, 2, 3, 5, 6 & 7 subunits monoclonal antibody (MCP231) 1:1,000 | Enzo | BML-PW8195 |
| Alexa-Fluor 488 1:400 | Invitrogen | A11001 |
| Alexa-Fluor 488 1:400 | Invitrogen | A11008 |
| **Chemicals, enzymes and other reagents** | | |
| Bortezomib | Sigma Aldrich | 5043140001 |
| Carfilzomib | Selleck Chemicals | S2853 |
| MG-132 | Sigma Aldrich | 474787 |
| Marizomib | Sigma Aldrich | SML1916 |
| MG132 | Enzo Life Sciences | BML-PI102-0005 |
| ProteoStat Aggresome Detection Kit | Enzo Life Sciences Inc. | ENZ-51035-K100 |
| LysC | Wako | 125-05061 |
| Trypsin | Promega | V5111 |
| Suc-LLVY-AMC | UBPBio | J4120 |
| Boc-LRR-AMC | UBPBio | J4120 |
| Z-LLE-AMC | UBPBio | J4120 |
| iRT kit | Biognosys AG | Ki-3002 |
| **Software** | | |
| STAR sequence aligner | Dobin *et al* (2013) | v2.7.1a |
| RSEM | Li and Dewey (2011) | v1.3.1 |
| Proteome Discoverer | Thermo Fisher Scientific | v2.0 |
| Xcalibur | Thermo Fisher Scientific | v4.0 |
| MaxQuant | https://www.maxquant.org/ | v1.5.3.28 |
| Spectronaut Professional+ | https://biognosys.com/shop/spectronaut | v12.0.20491.0.21234 |
| GraphPad Prism | https://www.graphpad.com/ | v7 and v8 |
| R Studio | https://www.r-project.org/ | |
| **Other** | | |
| Orbitrap Fusion Lumos | Thermo Fisher Scientific | |
| Q Exactive HF-X | Thermo Fisher Scientific | |

## Methods and Protocols

### Experimental animals including strain and husbandry details
#### Fish maintenance
The *N. furzeri* strain was maintained at the FLI facility as described in Baumgart *et al* (2014). To avoid effects of circadian rhythms and feeding, animals were always sacrificed at 10 am in fasted state. For tissue preparation, fish were euthanized with MS-222 (PharmaQ) and cooled on crushed ice. Animals used for *in vivo* pharmacological interventions (section *In vivo* proteasome inhibition) were euthanized by rapid-chilling. A complete list of fish used for the different experiments and is reported in Dataset EV1.

#### Mouse maintenance
All mice were C57BL/6JRj obtained from Janvier Labs or from internal breeding. Animals were maintained at the Leibniz Institute on Aging-Fritz Lipmann Institute (FLI) in a specific pathogen-free animal facility with a 12-h light/dark cycle. Mice were euthanized with $CO_2$. A complete list of mice used for the different experiments is reported in Dataset EV1.

#### Ethical statements
All experiments were performed in accordance with relevant guidelines and regulations. Fish were bred and kept in FLI's fish facility according to §11 of German Animal Welfare Act under license number J-003798. The protocols of animal experimentations were approved by the local authority in the State of Thuringia (Veterinaer- und Lebensmittelueberwachungsamt; proteasome inhibition: reference number 22-2684-04-FLI-19-010). Sacrifice and organ harvesting of non-experimental animals were performed according to §4(3) of German Animal Welfare Act.

### Size-exclusion chromatography coupled to mass spectrometry
#### Sample preparation
Individual brains of young (5 wph) and old (39 wph) fish were collected and snap-frozen in liquid nitrogen (Dataset EV1). On the preparation for size-exclusion chromatography (SEC), at least four brains were pooled for each replicate in order to obtain ~ 3 mg of protein extract as starting material and lysed in 1.5 ml of lysis buffer (50 mM HEPES, pH 6.8, 1 mM $MgCl_2$, 1 mM DTT, 150 mM NaCl, 5 mM ATP, cycloheximide 100 μg/ml, RNAse inhibitor 50 U, protease and phosphatase inhibitors). Samples were then vortexed (five times) prior to sonication using a Bioruptor Plus (Diagenode) for five cycles (30 s ON/60 s OFF) at high setting, at 4°C. The samples were then clarified by subsequent centrifugation steps as follows: (i) 500 *g* for 5 min at 4°C, (ii) 1,000 *g* for 13 min at 4°C, and (iii) 100,000 *g* for 30 min at 4°C. The final supernatant was concentrated using 30 kD spin filters (Merck Amicon Ultra −0.5 ml, centrifugal filters, UFC503096) to a final concentration of 10 μg/μl, as judged by OD280, and subjected to SEC as indicate below. Experiments were performed in duplicates for each age group and conducted in different days.

#### Size-exclusion chromatography
Size-exclusion chromatography was performed using an ÄKTA avant system equipped with UV detection at 280 nm wavelength. The column was a Yarra-SEC-4000 column (300 × 7.8 mm, pore size 500 Å, particle size 3 μm) with a SecurityGard™ cartridge

GFC4000 4 × 3.0 mm ID as a guard column. Running conditions were temperature 4°C, flow rate 0.5 ml/min, and run time of 40 min, and mobile phase was 50 mM HEPES, pH 6.8, 1 mM $MgCl_2$, 1 mM DTT, 150 mM NaCl, and 5 mM ATP. A standard sample (Phenomenex, ALO-3042) was injected prior to each sample to verify column performance. Sample amounts of 100 μl of a 10 mg/ml lysate were injected, corresponding to 1 mg protein extract on column. Fractions (200 μl each) were collected along with the LC separation directly in SDS buffer, to a final concentration of 4%. Thirty-nine fractions were further processed for LC-MS/MS analysis (see section Sample preparation for SEC fractions for sample preparation and section Data independent acquisition for SEC fractions for data acquisition).

### In vivo proteasome inhibition
Adult animals (12–14 wph) were subjected twice to pharmacological intervention via intraperitoneal injections (IP) during a 4-day period treatment. On the first and third day of the experiment ($t = 0$ and $t = 48$ h), fish were anesthetized with 200 mg/l buffered MS-222 (PharmaQ) and gently manipulated to deliver IP of either different drugs at 500 μM or vehicle (DMSO) at a dosage of 10 μl/g body weight. Animals from the same hatch were randomly allocated to the experimental groups. Both male and female fish were included in each experimental group. Adverse events were observed for some of the marizomib-treated animals as reported in Dataset EV1. After the fourth day of treatment ($t = 96$ h), fish were euthanized as previously described (Fish maintenance), brains harvested and used either for proteasome activity assay (Proteasome peptidase assay) or sample preparation for mass spectrometry (section Sample preparation for proteome analysis (*Nothobranchius furzeri*)). Different compounds were tested (see list above) in order to reach an optimal reduction of proteasome activity similar to the levels observed in old animals.

### Sample preparation for mass spectrometry analysis and data acquisition
#### Sample preparation for proteome analysis (Nothobranchius furzeri)
Individual brains from the fish were collected and snap-frozen in liquid nitrogen (Dataset EV1). On preparation for MS, protein amount was estimated based on fresh tissue weight (assuming 5% of protein w/w) and lysis buffer (4% SDS, 100 mM HEPES, pH8, 1 mM EDTA, 100 mM DTT) was added accordingly to a final concentration of 1 μg/μl. Samples were then vortexed (five times) prior to sonication (Bioruptor Plus) for 10 cycles (30 s ON/60 s OFF) at high setting, at 4°C. The samples were then centrifuged at 3,000 *g* for 5 min at room temperature, and the supernatant transferred to 2-ml Eppendorf tubes. Reduction (15 min, 45°C) was followed by alkylation with 20 mM iodoacetamide (IAA) for 30 min at room temperature in the dark. Protein amounts were confirmed, following an SDS–PAGE gel of 4% of each sample against an in-house cell lysate of known quantity. Between 200 and 300 μg of each sample was taken along for digestion. Proteins were precipitated overnight at −20°C after addition of a 4× volume of ice-cold acetone. The following day, the samples were centrifuged at 20,800 *g* for 30 min at 4°C and the supernatant carefully removed. Pellets were washed twice with 1 ml ice-cold 80% (v/v) acetone in water then centrifuged at 20,800 *g* at 4°C. They were then allowed to air-dry before addition of 120 μl of digestion buffer (3 M urea,

100 mM HEPES, pH8). Samples were resuspended with sonication (as above), LysC (Wako) was added at 1:100 (w/w) enzyme:protein, and digestion proceeded for 4 h at 37°C with shaking (Eppendorf ThermoMixer®C, thermoblock for 1.5 ml tubes, at 1,000 rpm for 1 h, then 650 rpm). Samples were then diluted 1:1 with Milli-Q water, and trypsin (Promega) added at the same enzyme to protein ratio. Samples were further digested overnight at 37°C with shaking (650 rpm). The following day, digests were acidified by the addition of TFA to a final concentration of 2% (v/v) and then desalted with Waters Oasis® HLB µElution Plate 30 µm (Waters Corporation, Milford, MA, USA) in the presence of a slow vacuum. In this process, the columns were conditioned with $3 \times 100$ µl solvent B (80% (v/v) acetonitrile; 0.05% (v/v) formic acid) and equilibrated with $3 \times 100$ µl solvent A (0.05% (v/v) formic acid in Milli-Q water). The samples were loaded, washed 3 times with 100 µl solvent A, and then eluted into 0.2-ml PCR tubes with 50 µl solvent B. The eluates were dried down with the speed vacuum centrifuge and dissolved at a concentration of 1 µg/µl in reconstitution buffer (5% (v/v) acetonitrile, 0.1% (v/v) formic acid in Milli-Q water). Reconstituted peptides were either analyzed directly (label-free analysis, see section Data acquisition for label-free analysis, used for TMT labeling, see section TMT labeling, or DIA, see section Data independent acquisition for In vivo proteasome inhibition).

### Sample preparation for SEC fractions

Additional DTT (to a final concentration of 50 mM) in 100 mM HEPES, pH 8 was added to each fraction, followed by sonication (Bioruptor Plus) for 10 cycles (30 s ON/60 s OFF) at high setting, at 20°C. Reduction and alkylation were performed as previously described (section Sample preparation for proteome analysis (Nothobranchius furzeri)). Protein amounts were estimated following an SDS–PAGE gel of 4% of each sample against an in-house cell lysate of known quantity. Between 10 and 40 µg of each fraction were taken along for digestion. Proteins were digested and peptide desalted as described in section Sample preparation for proteome analysis (Nothobranchius furzeri) and analyzed as described in section Data independent acquisition for SEC fractions.

### Isolation of SDS-insoluble protein aggregates

Individual brains from young (5 months) and old animals (21 or 26 months, Dataset EV1; between 481 and 533 mg wet tissue weight, $n = 5$) were lysed at a protein concentration of 30 µg/µl in lysis buffer (4% SDS, 100 mM HEPES, pH 8, 1 mM EDTA, 100 mM DTT). Samples were then vortexed (five times) prior to sonication with a Bioruptor Plus (high setting, 20°C, 20 cycles of 60 s ON/30 s OFF). Samples were then centrifuged at 20,000 g for 5 min at room temperature, and the supernatant transferred to fresh 2-ml Eppendorf tubes (total homogenate; TH). In order to obtain pellets of aggregates, 200 µl of brain lysates (TH) was transferred to polypropylene thick wall tubes (Beckman Coulter), in duplicate, and submitted to ultracentrifugation at 100,000 g for 30 min at 20°C. Supernatant was transferred to a fresh tube (supernatant; SUP) and remaining pellet washed twice, by resuspension with 200 µl of lysis buffer, followed by ultracentrifugation. Supernatant from each of the washes was transferred to fresh tubes (wash 1; W1 and wash 2; W2). In order to facilitate their solubilization, pellets (SDS-insoluble proteins) were then submitted to 50 µl of neat formic acid for 1 h at 37°C with shaking (Eppendorf ThermoMixer® C, thermoblock for

1.5 ml tube, at 400 rpm). After incubation, samples were speed vacuum centrifuged at 45°C, resuspended in 50 µl of lysis buffer, and boiled at 95°C for 10 min. Samples were then transferred to 0.5-ml Eppendorf tubes and rebuffered with 5 M NaOH. To obtain protein aggregates of N. furzeri, same procedure was repeated for young (5 wph) and old (30 wph) animals (Dataset EV1), including minor protocol modifications. At least 3 animals were pooled for each replicated ($n = 5$) and lysed to 30 µg/µl. Because of the small brain size, only 30–120 µl lysate (TH) were obtained for replicate. An equal volume of TH across samples were kept for further mass spectrometry analysis and the remaining volume were submitted to ultracentrifugation. On preparation for MS, total homogenate (10 µg) and equal volumes of resuspended pellets (estimated amount of protein between 5 and 15 µg, for mice samples) for each sample were submitted to protein precipitation, digestion, and clean up as described in section Sample preparation for proteome analysis (Nothobranchius furzeri).

### Quantification of SDS-insoluble aggregates from young and old brains (Mus musculus)

Resuspended pellets were loaded in SDS–PAGE gel, and their protein content compared. In this process, equal volumes of resuspended/rebuffered pellets were mixed with 2× loading buffer (1.5 M Tris–pH 6.8, 20% SDS, 85% glycerin), loaded in precast protein gel (Bio-Rad, Mini-PROTEAN TGX 4–20%, 10-well), and run under constant mode (100 V, 1:30 h) in 1% SDS running buffer. The gel was then stained with Coomassie overnight at room temperature, with shaking, followed by extensive washing with Milli-Q water. The image was acquired using the ChemiDoc XRS+ system (Bio-Rad) with the standard colorimetric settings (Image Lab 5.2.1). A high resolution image was then exported for further densitometry analysis of full-length lanes with the open source software ImageJ 1.52a on a Windows 7 Professional 64-bit install (NIH) (Schneider et al, 2012). Prior to analysis, the image was converted to gray scale 8-bit mode. In brief, a rectangular selection of same size was drawn across the full lane of each sample. Profile plots with the relative density of the contents from each rectangle were generated (function Plot Lanes). The generated area under the curve for each sample was measured (function Wand). In order to test for significant differences between young and old brains, density area values were tested for normal distribution (Shapiro–Wilk test). An unpaired parametric t-test was performed. Statistical analysis was done using built-in functions of GraphPad Prism 8.

### Data acquisition for label-free analysis

Peptides were separated using the nanoAcquity UPLC system (Waters) fitted with a trapping (nanoAcquity Symmetry $C_{18}$, 5 µm, 180 µm × 20 mm) and an analytical column (nanoAcquity BEH $C_{18}$, 1.7 µm, 75 µm × 250 mm). The outlet of the analytical column was coupled directly to an Orbitrap Fusion Lumos (Thermo Fisher Scientific) using the Proxeon nanospray source. Solvent A was water, 0.1% (v/v) formic acid, and solvent B was acetonitrile, 0.1% (v/v) formic acid. The samples (500 ng) were loaded with a constant flow of solvent A at 5 µl/min onto the trapping column. Trapping time was 6 min. Peptides were eluted via the analytical column with a constant flow of 0.3 µl/min. During the elution step, the percentage of solvent B increased in a linear fashion from 3 to 25% in 30 min and then increased to 32% in five more minutes and finally to 50%

in a further 0.1 min. Total runtime was 60 min. The peptides were introduced into the mass spectrometer via a Pico-Tip Emitter 360 μm OD × 20 μm ID; 10 μm tip (New Objective), and a spray voltage of 2.2 kV was applied. The capillary temperature was set at 300°C. The RF lens was set to 30%. Full scan MS spectra with mass range 375–1,500 $m/z$ were acquired in profile mode in the Orbitrap with resolution of 120,000 FWHM. The filling time was set at maximum of 50 ms with limitation of $2 \times 10^5$ ions. The "Top Speed" method was employed to take the maximum number of precursor ions (with an intensity threshold of $5 \times 10^3$) from the full scan MS for fragmentation (using HCD collision energy, 30%) and quadrupole isolation (1.4 Da window) and measurement in the ion trap, with a cycle time of 3 s. The monoisotopic precursor selection () peptide algorithm was employed but with relaxed restrictions when too few precursors meeting the criteria were found. The fragmentation was performed after accumulation of $2 \times 10^3$ ions or after filling time of 300 ms for each precursor ion (whichever occurred first). MS/MS data were acquired in centroid mode, with the Rapid scan rate and a fixed first mass of 120 $m/z$. Only multiply charged ($2^+$– $7^+$) precursor ions were selected for MS/MS. Dynamic exclusion was employed with maximum retention period of 60 s and relative mass window of 10 ppm. Isotopes were excluded. Additionally, only one data-dependent scan was performed per precursor (only the most intense charge state selected). Ions were injected for all available parallelizable time. In order to improve the mass accuracy, a lock mass correction using a background ion ($m/z$ 445.12003) was applied. For data acquisition and processing of the raw data, Xcalibur 4.0 (Thermo Scientific) and Tune version 2.1 were employed.

### TMT labeling

The resuspended peptides (at 1 μg/μl) were rebuffered to pH 8.5 using 1 M HEPES prior labeling. For the total proteome experiment, 12 wph samples were used as common reference for 5 and 39 wph samples (see Dataset EV1 for labeling scheme). In this case, 30 μg and 15 μg of peptides were taken for each labeling reaction, from 12 and 5 wph/39 wph samples, respectively. For TPP experiments, 10 μg of peptides from each temperature point from each replicate was labeled (see Dataset EV1 for labeling scheme). TMT-10plex reagents (Thermo Scientific) were reconstituted in 41 μl of anhydrous DMSO. TMT labeling was performed in two steps by addition of 2× of the TMT reagent per μg of peptide (e.g., 60 μg of TMT reagent for 30 μg of peptides). First, sample amount of TMT reagent was added to samples at room temperature, with shaking at 600 rpm in a thermomixer (Eppendorf) for 30 min. After incubation, a second portion of TMT reagent was added and incubated for another 30 min. After checking labeling efficiency by MS, samples were pooled (50 μg total), desalted as described in Sample preparation for proteome analysis (*Nothobranchius furzeri*), and subjected to high pH fractionation prior to MS analysis.

### High pH peptide fractionation for TMT labeled samples

Offline high pH reverse phase fractionation was performed using an Agilent 1260 Infinity HPLC System equipped with a binary pump, degasser, variable wavelength UV detector (set to 220 and 254 nm), Peltier-cooled autosampler (set at 10°C), and a fraction collector. The column was a Waters XBridge C18 column (3.5 μm, 100 × 1.0 mm, Waters) with a Gemini C18, 4 × 2.0 mm SecurityGuard (Phenomenex) cartridge as a guard column. The solvent system consisted of

20 mM ammonium formate (pH 10.0) as mobile phase (A) and 100% acetonitrile as mobile phase (B). The separation was accomplished at a mobile phase flow rate of 0.1 ml/min using a non-linear gradient from 95 A to 40% B in 91 min. Forty-eight fractions were collected along with the LC separation that were subsequently pooled into 16 non-consecutive fractions. Pooled fractions were dried in a Speed-Vac and then stored at −80°C until LC-MS/MS analysis.

### Data acquisition TMT labeled, high pH fractionated samples

For TMT experiments, fractions were resuspended in 10 μl reconstitution buffer (5% (v/v) acetonitrile, 0.1% (v/v) TFA in water), and 3 μl was injected. Peptides were separated using the nanoAcquity UPLC system (Waters) fitted with a trapping (nanoAcquity Symmetry C18, 5 μm, 180 μm × 20 mm) and an analytical column (nanoAcquity BEH C18, 2.5 μm, 75 μm × 250 mm). The outlet of the analytical column was coupled directly to an Orbitrap Fusion Lumos (Thermo Fisher Scientific) using the Proxeon nanospray source. Solvent A was water, 0.1% (v/v) formic acid, and solvent B was acetonitrile, 0.1% (v/v) formic acid. The samples were loaded with a constant flow of solvent A at 5 μl/min, onto the trapping column. Trapping time was 6 min. Peptides were eluted via the analytical column at a constant flow of 0.3 μl/min, at 40°C. During the elution step, the percentage of solvent B increased in a linear fashion from 5 to 7% in 10 min and then from 7 B to 30% B in a further 105 min and to 45% B by 130 min. The peptides were introduced into the mass spectrometer via a Pico-Tip Emitter 360 μm OD × 20 μm ID; 10 μm tip (New Objective), and a spray voltage of 2.2 kV was applied. The capillary temperature was set at 300°C. Full scan MS spectra with mass range 375–1,500 $m/z$ were acquired in profile mode in the Orbitrap with resolution of 60,000 FWHM using the quadrupole isolation. The RF on the ion funnel was set to 40%. The filling time was set at maximum of 100 ms with an AGC target of $4 \times 10^5$ ions and 1 microscan. The peptide MIPS was enabled along with relaxed restrictions if too few precursors were found. The most intense ions (instrument operated for a 3 s cycle time) from the full scan MS were selected for MS2, using quadrupole isolation and a window of 1 Da. HCD was performed with collision energy of 35%. A maximum fill time of 50 ms for each precursor ion was set. MS2 data were acquired with fixed first mass of 120 $m/z$ in the ion trap. The dynamic exclusion list was with a maximum retention period of 60 s and relative mass window of 10 ppm. The instrument was not set to inject ions for all available parallelizable time. For the MS3, the precursor selection window was set to the range 400–2,000 $m/z$, with an exclude width of 18 $m/z$ (high) and 5 $m/z$ (low). The most intense fragments from the MS2 experiment were co-isolated (using synchronous precursor selection = 8) and fragmented using HCD (65%). MS3 spectra were acquired in the Orbitrap over the mass range 100–1,000 $m/z$ and resolution set to 30,000 FWHM. The maximum injection time was set to 105 ms, and the instrument was set not to inject ions for all available parallelizable time.

### Data independent acquisition for *in vivo* proteasome inhibition

Reconstituted peptides were spiked with retention time iRT kit (Biognosys AG, Schlieren, Switzerland). Peptides were separated using the nanoAcquity UPLC system (Waters) with a trapping (nanoAcquity Symmetry C18, 5 μm, 180 μm × 20 mm) and an analytical column (nanoAcquity BEH C18, 1.7 μm, 75 μm × 250 mm). The outlet of the column was coupled to a Q

exactive HF-X (Thermo Fisher Scientific) using the Proxeon nanos-pray source. Solvent A was water, 0.1% FA, and solvent B was acetonitrile, 0.1% FA. Samples were loaded at constant flow of solvent A at 5 μl/min onto the trap for 6 min. Peptides were eluted via the analytical column at 0.3 μl/min and introduced via a Pico-Tip Emitter 360 μm OD × 20 μm ID; 10 μm tip (New Objective). A spray voltage of 2.2 kV was used. During the elution step, the percentage of solvent B increased in a non-linear fashion from 0 to 40% in 120 min. Total run time was 145 min. The capillary temperature was set at 300°C. The RF lens was set to 40%. MS conditions were as follows: Full scan MS spectra with mass range 350–1,650 $m/z$ were acquired in profile mode in the Orbitrap with resolution of 120,000 FWHM. The filling time was set at maximum of 60 ms with limitation of $3 \times 10^6$ ions. DIA scans were acquired with 40 mass window segments of differing widths across the MS1 mass range. The default charge state was set to 3+. HCD fragmentation (stepped normalized collision energy; 25.5, 27, 30%) was applied, and MS/MS spectra were acquired with a resolution of 30,000 FWHM with a fixed first mass of 200 $m/z$ after accumulation of $3 \times 10^6$ ions or after filling time of 35 ms (whichever occurred first). Data were acquired in profile mode. For data acquisition and processing of the raw data, Xcalibur 4.0 (Thermo Scientific) and Tune version 2.9 were employed.

### Data independent acquisition for SEC fractions

Data acquisition was performed as described above, including modifications in the gradient settings and MS as follows. During the elution step, the percentage of solvent B increased in a non-linear fashion from 0 to 40% in 60 min. Total run time was 75 min. DIA scans were acquired with 30 mass window segments. HCD fragmentation (stepped normalized collision energy; 25.5, 27, 30%) was applied, and MS/MS spectra were acquired with a resolution of 30,000 FWHM with a fixed first mass of 200 $m/z$ after accumulation of $3 \times 10^6$ ions or after filling time of 47 ms (whichever occurred first).

### Data processing for TMT labeled samples (total proteome analysis)

TMT-10plex data were processed using Proteome Discoverer v2.0 (Thermo Fisher Scientific). Data were searched against the relevant species-specific fasta database (in-house *N. furzeri* or UniProt database, SwissProt entry only, release 2016_01 *for mouse* or UniProt database, SwissProt entry only, release 2016_01 *for human*) using Mascot v2.5.1 (Matrix Science) with the following settings: Enzyme was set to trypsin, with up to 1 missed cleavage. MS1 mass tolerance was set to 10 ppm and MS2 to 0.5 Da. Carbamidomethyl cysteine was set as a fixed modification and oxidation of methionine as variable. Other modifications included the TMT-10plex modifications from the quantification method used. The quantification method was set for reporter ions quantification with HCD and MS3 (mass tolerance, 20 ppm). The false discovery rate for peptide-spectrum matches (PSMs) was set to 0.01 using Percolator (Brosch *et al*, 2009).

Reporter ion intensity values for the PSMs were exported and processed with procedures written in R (version 3.5.0) using R-studio (version 1.0.153), as described in Heinze *et al* (2018). Briefly, PSMs mapping to reverse or contaminant hits, or having a Mascot score below 15, or having reporter ion intensities below $1 \times 10^3$ in all the relevant TMT channels were discarded. TMT channels intensities from the retained PSMs were then $\log_2$ transformed, normalized, and summarized into protein group quantities by taking the

median value. At least two unique peptides per protein were required for the identification, and only those peptides with no missing values across all 10 channels were considered for quantification. Protein differential expression was evaluated using the limma package (Ritchie *et al*, 2015). Differences in protein abundances were statistically determined using Student's *t*-test moderated by the empirical Bayes method. *P* values were adjusted for multiple testing using the Benjamini–Hochberg method (FDR, denoted as "adj. *P*") (Benjamini & Hochberg, 1995). The results are reported in Dataset EV2.

To obtain iBAQ values (Schwanhäusser *et al*, 2011) from TMT data, all samples were re-analyzed with MaxQuant 1.5.3.28. The parameters were set identically to the analyses above. For each identified peptide in each separate LC-MS analysis file, we extracted both precursor intensities and corresponding PSMs from the evidence files. For further analysis, we only considered PSMs that were common to both the TMT quantification approach as described above and the MaxQuant analysis and were filtered as before. For each resulting peptide per LC-MS analysis, we calculated a peptide ratio corresponding to the median of the ratios derived from its PSMs. Given that the total area (MS1/precursor intensity) of a peptide species represents the sum of the 10 TMT channels, splitting the total area into individual channels using the TMT ratios gives the intensity portions for each channel. After splitting, we corrected for potential sampling aberrations by multiplying the area intensities per channel with the median ratio determined from the TMT ratios. For each channel and peptide, we calculated label-free scores by dividing through the number of potentially observable unique tryptic peptides per protein (criteria: peptide length 8–25 amino acids, no missed cleavage allowed). The resulting iBAQ scores of unique peptides were summed up per protein, and protein scores were normalized across samples using median normalization.

### Data processing for label-free quantification (protein aggregates)

Software MaxQuant (version 1.5.3.28) was used to search the data. The data were searched against a species-specific (*N. furzeri* in-house or UniProt database, SwissProt entry only, release 2016_01 for mouse) database with a list of common contaminants appended. The data were searched with the following modifications: Carbamidomethyl (C) (fixed) and Oxidation (M) and Acetyl (Protein N-term; variable). The mass error tolerance for the full scan MS spectra was set at 20 ppm and for the MS/MS spectra at 0.5 Da. A maximum of two missed cleavages was allowed. For aggregate analysis, iBAQ values (Schwanhäusser *et al*, 2011) from the MaxQuant output were used to perform quantitative analyses. Only protein groups quantified in at least two replicates per sample group were retained. To reduce technical variation, data were $\log_2$ transformed and quantile normalized using the preprocessCore library. Differential protein expression was assessed using the limma package, as described in section Data processing for TMT labeled samples (total proteome analysis). The results are reported in Dataset EV7.

### Data processing for DIA

For experiment-specific library creation, the DIA data were searched against either a *N. furzeri* in-house database (59,154 entries) or a *N. furzeri* UniProt database (35,275 entries) and a list of common contaminants using Pulsar engine in Spectronaut Professional+

(version 12.0.20491.0.21234, Biognosys AG, Schlieren, Switzerland). The following modifications were included in the search: Carbamidomethyl (C) (Fixed) and Oxidation (M)/Acetyl (Protein N-term; Variable). A maximum of 2 missed cleavages for trypsin were allowed. The identifications were filtered to satisfy FDR of 1% on peptide and protein level.

For the SEC experiment (see section Size-exclusion chromatography coupled to mass spectrometry), a library containing all samples (5 wph and 39 wph, in duplicates) was generated and contained 117,755 precursors. The same library was used to search data from different experiments separately. Precursor matching, protein inference, and quantification were performed in Spectronaut using default settings.

For the *in vivo* proteasome inhibition experiment (see section *In vivo* proteasome inhibition), the library generated from all DIA data (DMSO and Bortezomib) contained 70,301 precursors, corresponding to 6.636 protein groups.

The protein quantity report was then exported, and further data analyses and visualization were performed with R using in-house pipelines and scripts. For *in vivo* proteasome inhibition experiment, differential protein expression was assessed using the limma package, as described in section Data processing for TMT labeled samples (total proteome analysis). SEC analysis was performed as described in section Analysis of SEC-MS data.

### Sample preparation and data processing for RNA sequencing
#### RNA isolation
RNA from each sample was extracted from protein lysates using QIAzol lysis reagent (Qiagen). In brief, 1 ml of QIAzol reagent was added to 100 μl of lysate, followed by the addition of 200 μl of chloroform. Samples were mixed vigorously and centrifuged at 12,000 *g* for 20 min at 4°C, after 3-min incubation at room temperature. The upper aqueous phase was carefully transferred into a fresh tube and mixed with one volume of isopropyl alcohol, 0.16 volumes of sodium acetate (2 M; pH 4.0), and 1 μl of GlycoBlue (Invitrogen™) in order to precipitate RNA. After 10-min incubation at room temperature, samples were centrifuged at 12,000 *g* for 30 min at 4°C. The supernatant was completely removed, and RNA pellets were washed by adding 80% (v/v) ethanol and centrifuging at 7,500 *g* for 5 min at 4°C. The washing steps were performed twice. The resulting pellets were air-dried for no more than 5 min and dissolved in 10 μl nuclease-free water. To ensure full dissolution of RNA in water, samples were then incubated at 65°C for 5 min, before storage at −80°C.

#### Library preparation
Sequencing of RNA samples was done using Illumina's next-generation sequencing methodology (Bentley *et al*, 2008). In detail, quality check and quantification of total RNA were done using the Agilent Bioanalyzer 2100 in combination with the RNA 6000 pico kit (Agilent Technologies). Total RNA library preparation was done introducing 250 ng total RNA into the Illumina's TruSeq Stranded Total RNA Library Prep Kit/RiboZero Gold kit, following the manufacturer's instructions. Small RNA library preparation was done using Illumina's TruSeq small RNA library preparation kit following the manufacturer's description. Quality and quantity of all libraries were checked using Agilent's Bioanalyzer 2100 and DNA 7500 kits.

#### Sequencing
All libraries were sequenced on a HiSeq2500 running in 51 cycle/single-end/high-output mode (sequencing chemistry v3). Total RNA libraries were pooled and sequenced in three lanes. Small RNA libraries were pooled and sequenced in one lane. Sequence information was extracted in FastQ format using Illumina's bcl2fastq v.1.8.4. Sequencing of total RNA libraries resulted in around 42 mio reads per sample and sequencing of small RNA libraries in around 12 mio reads per sample.

#### Data processing for small RNAs
Sequence information was extracted in FastQ format using Illumina's bcl2fastq software v1.8.3. The processing and annotation of small RNA-seq raw data were performed using the R programming language (version 3.0.2) and the ShortRead Bioconductor package (Morgan *et al*, 2009). First, raw data were preprocessed with the following parameters: Quality filtering, eliminating all reads containing an "N"; Adapter trimming, by use of the function trimLRPatterns(), allowing up to two mismatches and using as adapter sequence "TGGAATTCTCGGGTGCCAAGGAACTCCAGTCAC". Size filtering removed all the reads with lengths shorter than 18 and longer than 33 nucleotides. Reads were aligned using Bowtie 1.1.2 (Langmead *et al*, 2009), resulting in a direct annotation and quantification. The alignment was divided in two steps, to allow the recognition and the annotation of the reads exceeding reference length. First, we performed alignment against the reference (mature miRNAs, *N. furzeri* reference catalogue (Baumgart *et al*, 2017)) with up to two mismatches. In this step, the reference used was the mature sequence of microRNAs. Each read was aligned using these criteria with Bowtie (settings: "-q", "–threads 8 –best", "—norc"). The remaining reads, which could not be aligned in the previous step, were used as reference for a second alignment step with Bowtie 1.1.2 (settings: -f", "-a", "–threads 8 –norc"). In this case, the annotated mature microRNAs were aligned against the reads. The information obtained in the two alignment phases was conveyed in one single Dataset, containing a list of all the retrieved sequences and their relative counts.

RNA-seq data were then processed as follow: Sequences were mapped using Tophat2 (-T -x 1) (Kim *et al*, 2013) to the Nfu_20150522 genome. Counting was performed using featureCounts -s 0 on the genebuild_v1.150922 *N. furzeri* annotation. For both small and coding RNAs, the raw counts were analyzed with DESeq2 package (Love *et al*, 2014) for differential expression. Differential expression was performed independently for the two comparisons shown in the work (12 vs 5 wph and 39 vs 12 wph) with $\alpha = 0.05$. The analysis was applied to both coding and non-coding RNAs (miRNA-seq data). For miRNA analysis, only miRNAs up-regulated with aging ($\log_2$ fold change $> 0$ and adj. $P < 0.05$) were considered. The results are reported in Dataset EV3.

#### Longitudinal study
*Nothobranchius furzeri* longitudinal data were obtained from: (i) an already published data set (GSE66712, 90 longitudinal fin-clip datasets from 45 animals, (Baumgart *et al*, 2016)), and (ii) newly sequenced 228 fin-clip samples from additional 114 animals of the same cohort. Sample preparation, sequencing, and data analysis were done as described in Baumgart *et al* (2014, 2016).

### Ribosome footprinting

RNA from additional samples (young, 6 wph; old, 26 wph; four replicates each age group) were extracted and sequenced as previously described (section RNA isolation and Sequencing). Libraries were prepared following manufacturer's instructions without depletion of ribosomal RNA (ARTseq, Epicentre).

## Measurements of proteasome activity

### Native gel electrophoresis and in-gel proteasome assay

Brains from young, adult, and old fish were lysed by sonication using a Bioruptor for five cycles (30 s ON/60 s OFF) at high setting, at 4°C, in proteasome activity lysis buffer (50 mM Tris–HCl, pH 7.4, 5 mM MgCl$_2$, 5 mM ATP, 1 mM DTT, 10% glycerol). Cell lysates were then centrifuged at 16,000 $g$ for 10 min, at 4°C. Protein concentration was determined using the Bradford method with bovine serum albumin as standard. Forty microgram of total cell lysates was subjected to native gel electrophoresis to reveal the various proteasome complexes (30S, 26S, 20S). The gel was then incubated in 50 mM Tris, pH 7.4, 5 mM MgCl$_2$, 1 mM ATP, and 300 μM proteasome substrate (Suc-LLVY-AMC; UBPBio) for 30 min at 37°C to assay chymotrypsin-like (CT-L) activity. Proteasome bands were visualized under UV. Following the CT-L activity assay, proteins in native gels were transferred to nitrocellulose membranes for immunoblotting with a monoclonal antibody against the α1, 2, 3, 5, 6, and 7 members of the 20S proteasome subunit (MCP231, Enzo BML-PW8195, 1:1,000 dilution in TBS (0.1% Tween) and 5% milk). Equal protein loading was confirmed by Ponceau staining and immunoblotting with α-tubulin antibody. Briefly, 20 μg of the same total cell lysates used for loading native gels was then fractionated by SDS–PAGE and transferred to nitrocellulose membranes for probing with a monoclonal antibody against α-tubulin (T9026, Sigma, 1:5,000). Secondary antibodies used were conjugated with horseradish peroxidase, and Clarity™ Western ECL substrate (Bio-Rad Laboratories, Hercules, USA) was used to develop the blots.

### Proteasome peptidase assay

Brains from young, adult, and old animals were sonicated as described in 6.1. CT-L, T-L, and PGPH proteasome activities were assayed with the hydrolysis of specific fluorogenic peptides (UBPBio), namely Suc-LLVY-AMC (for CT-L activity), Boc-LRR-AMC (for T-L activity), and Z-LLE-AMC (for PGPH activity), respectively, for 1 h at 37°C. 10 μg of total cell lysates was incubated in 50 mM Tris–HCl, pH 7.4, 5 mM MgCl$_2$, 1 mM ATP, 1 mM DTT, 10% glycerol, and 10 μM proteasome substrate for 1 h at 37°C. Specific proteasome activity was determined as the difference between the total activity of protein extracts and the remaining activity in the presence of 20 μM MG132 (Enzo Life Sciences). Fluorescence was measured by multiple reads for 60 min at 37°C by TECAN Kinetic Analysis (excitation 380 nm, emission 460 nm, read interval 5 min) on a Safire II microplate reader (TECAN).

## Immunofluorescence

The whole brains of young (7–10 wph) and old (27–30 wph) fish were fixed overnight using a solution of paraformaldehyde (PFA) 4% in phosphate buffer (PB) 100 mM and cryoprotected with a solution of Sucrose 30% for 24 h. Finally, tissues were included in OCT embedding medium (Tissue-tek, Sakura) and stored at −20°C until use. Brain sections (20 μm) were cut using a Leica cryostat, collected on Superfrost Plus slides (Menzel-Glaeser), and dried at 37°C for 2 h and then washed in PBS (three washings for 5 min each) to remove the embedding medium followed by an acid antigen retrieval treatment (10 mM Tri-sodium citrate, 0.05% Tween, pH 6). The solution was brought to the boiling point in a microwave, and the slides were dipped in the hot solution for 1 min, three times.

To visualize protein aggregates (Figs 3E and EV4E–H), an aggresome fluorescent staining kit that binds to the beta-sheets of protein aggregates was applied. Following the protocol described from Shen et al (2011), a solution 1:2,000 of aggresome dye in PBS was used for 3 min, followed by washing in PBS (three washings for 5 min each), and the sections were immersed in a solution of 1% acetic acid for 30 min at room temperature to remove the excess of staining (destaining step). This step was followed by an immunofluorescence procedure: Blocking solution (5% (w/v) BSA, 0.3% (v/v) Triton-X in PBS) was applied for 2 h at room temperature (RT), then the primary antibodies at the specific dilution in a solution of 1% (w/v) BSA, 0.1% (v/v) Triton in PBS, and the samples left overnight at 4°C. The following day, sections were washed in PBS (three washings for 5 min each) to eliminate the primary antibodies solutions, and the secondary antibody was applied (Alexa fluor 488, working dilution 1:400) for 2 h at RT. Samples were then washed with PBS (three washings for 5 min each) and mounted with a fluorescence mounting medium supplemented with DAPI as a nuclear staining (DAPI-Fluoroshield). Images were collected at different magnifications with a Zeiss Apotome.2 microscope provided with a ZEN 2 Pro software for the image processing and saved as TIFF format.

To detect age-dependent gliosis (Fig EV2B), double staining for S100 and glial fibrillary acidic protein (GFAP) was performed, following the standard immunohistochemistry protocols already described above: The specific primary antibodies were incubated simultaneously overnight at 4°C, followed by PBS washings and simultaneous incubation with the specific secondary antibodies for 2 h at room temperature.

## Data analysis

### Mapping of N. furzeri genes to human orthologues

In order to map *N. furzeri* genes to human orthologues, the *N. furzeri* proteome (Reichwald *et al*, 2015) was directly mapped against the *Homo sapiens* UniProt reference proteome (accessed July 2017) using BLASTp. The hit with highest alignment score was selected as the orthologue. This procedure made it possible to assign 13,484 orthologues. Teleost fish underwent whole-genome duplication after their lineage separated from the one that includes mammals; thus, paralogue fish genes might diverge from the correspondent mammalian orthologue (hidden orthology). In order to detect possible candidates, the *N. furzeri* proteome was aligned against the spotted gar (*Lepisosteus oculatus*) proteome, and these orthologues consequently mapped on the human proteome. Spotted gar is a fish whose lineage diverged from teleosts before their genome duplication; thus, its genome could be used as a bridge to identify hidden orthologues between teleosts and mammals (Braasch *et al*, 2016). The procedure resulted in the assignment of further 146 (hidden) orthologues, equivalent to 1.1% of total orthologues mapped. The mapping table is reported in Dataset EV1.

### Transcriptome/proteome comparison

For the comparison between transcriptome and proteome two independent analyses were performed, one comparing 12 vs 5 wph brain samples, and the other one comparing 39 vs 12 wph samples. Differentially expressed genes and proteins were obtained for these and then plotted: For the visualization of the comparison (Fig 1D and E), only genes differentially expressed in both the comparisons (adj. $P < 0.05$) were considered. To identify enriched KEGG pathways (Fig 1G and H), a metanalysis was performed on the two dataset: $P$ values were combined using Fisher's method and then adjusted for multiple testing using the Benjamini–Hochberg method (Benjamini & Hochberg, 1995). For the gene enrichment, genes contained in each quadrant were analyzed separately with WebGestalt using KEGG pathways with a cut-off of FDR < 0.05. To obtain the correlation between protein and transcript levels for the different time points, normalized counts (obtained from DESeq2 package) and IBAQ (obtained from MaxQuant) were, respectively, used. Pearson correlation was calculated for every sample pair (Fig 1C).

### Analysis of SEC-MS data

The same complex definitions used for stoichiometry analysis were used (Dataset EV6). Protein quantification report exported from Spectronaut (see section Data processing for DIA) was processed using R 3.5.3, using in-house generated functions. Intensity values for each protein across the 39 fractions were normalized, dividing each value by the total sum of intensities along the fractions. Only complexes with at five subunits identified were considered for further analysis. For each experiment, the distribution of pairwise Pearson's correlation between members of the same protein complex was analyzed. A set of randomly defined protein complexes of same length was generated as a control for this analysis. For each complex, the median abundance of all the quantified subunits was used to generate the complex profile across fractions. All the complex profiles were scaled to the max value (set to 1) to make profiles comparable across experiments. A heatmap of the different protein complexes across the fractions was generated using the R package *pheatmap*. The results of SEC-MS data are reported in Dataset EV5.

### Protein complex analysis

Protein complex stoichiometries were analyzed using method and complex definitions described in Ori *et al* (2016). Briefly, protein intensities obtained from TMT experiments were assigned to the respective protein complex and normalized by the mean complex abundance (estimated using the trimmed mean protein intensity of all complex members). In order to identify stoichiometry changes, the complex-normalized protein matrix was used for differential expression analysis using limma as described in section Data processing for TMT labeled samples (total proteome analysis). Only protein complexes that had at least five members quantified were retained for analysis. Protein complexes were considered as "affected" if they had at least two members that showed differential expression across the conditions tested (adj. $P < 0.05$ and absolute $\log_2$ fold change > 0.5), unless otherwise stated. The results are reported in Dataset EV6 for *N. furzeri* aging and Dataset EV8 for *in vivo* proteasome inhibition experiment.

### Calculation of biophysical properties for aggregates

Predictions of biophysical properties from the amino acid sequence of proteins in the dataset were computed using the cleverSuite (Klus

*et al*, 2014) (chaperone requirement, intrinsic disorder) and s2D (Sormanni *et al*, 2015) (intrinsic disorder) classifiers. A biophysical property score is then assigned to proteins from the output of the classifier. CleverSuite is trained on a dataset of proteins that show the property under study (e.g., proteins requiring chaperons to fold, positive database) vs a "negative" database (e.g., self-folding proteins). The minimal output of the classifier is a label according to the similarity to one of the two datasets (positive, negative, or indeterminate) and the associated probability $P$ of correct classification, that can be converted to the score:

$$s_{bp}^{cS} = \begin{cases} -1 \cdot P(s_{bp} = x) & \text{if negative} \\ 0 & \text{if indeterminate} \\ P(s_{bp} = x) & \text{if positive} \end{cases}.$$

The s2D classifier determines the likelihood that a residue in a protein sequence would be included in an α-helix, β-sheet, or random coil; its minimal output is a string of predicted secondary conformations for each amino acid of the protein. A global score of disorder could be then determined from the number of residues predicted in a random coil state as follows:

$$s_c^{s2D} = \frac{n_c}{N}$$

where $n_c$ is the number of residues predicted to assume a random coil conformation, and $N$ is the total number of residues. Analysis of correlation between predicted scores and $T_m$ or protein enrichment in aggregates was performed using custom Python 3.5 scripts. Statistical analysis was done using built-in functions of GraphPad Prism 7.

### Analysis of N. furzeri ribosome footprinting data

Ribosome footprinting data were generated from young and aged killifish brain extracts (6 and 26 wph, respectively, see section Ribosome footprinting). Footprints were mapped to the *N. furzeri* reference genome (NotFur1) using the STAR sequence aligner (version 2.7.1a, (Dobin *et al*, 2013)). The relative abundance of footprints per gene was quantified with RSEM (version 1.3.1, (Li & Dewey, 2011)).

### Cox–Hazard model for longitudinal RNA-seq data

In order to isolate predictive factors of aging, survival analysis was performed to correlate mortality risk to gene expression. Cox–Hazard model (also called Cox regression) was applied using the formula:

$$h(t|\Delta_{ij}) = h_0(t) \exp(c_i \Delta_{ij})$$

$$\Delta_{ij} = \log_2(g_{ij(20)}/g_{ij(10)})$$

were $h_0(t)$ is the baseline hazard function, $g_{ij(20)}$ is the expression of gene $i$ in the sample $j$ at age 20 weeks, $g_{ij(10)}$ is the expression of gene $i$ in the sample $j$ at age 10 weeks and $c_i$ is a coefficient.

If $c_i > 0$, mortality increase the larger $\Delta_{ij}$, if $c_i < 0$, mortality decreases the larger $\Delta_{ij}$.

The analysis was performed using the *survival* package in R environment for all 159 animals that survived longer than 20 weeks. Normalized pseudo-counts obtained from DESeq2 were used as input, and the $c_i$ values were used as input for gene set enrichment using *gage* (Luo *et al*, 2009). The results of the Cox–Hazard analysis are reported in Dataset EV9.

# Data availability

The mass spectrometry proteomics data have been deposited to the ProteomeXchange Consortium via the PRIDE (Vizcaino *et al*, 2016) partner repository with the following dataset identifiers: PXD0123 14 (http://www.ebi.ac.uk/pride/archive/projects/PXD012314; TMT aging data), PXD018399 (http://www.ebi.ac.uk/pride/archive/projects/PXD018399; protein aggregate analysis), PXD016587 (http://www.ebi.ac.uk/pride/archive/projects/PXD016587; SEC), and PXD016459 (http://www.ebi.ac.uk/pride/archive/projects/PXD016459; proteasome inhibition experiments).

The sequencing data discussed in this publication have been deposited in NCBI's Gene Expression Omnibus and are accessible through GEO with the following dataset identifiers: GSE52462 (http://www.ncbi.nlm.nih.gov/geo/query/acc.cgi?acc = GSE52462; polyA$^+$ RNA-seq data from (Baumgart *et al*, 2014)), GSE125373 (http://www.ncbi.nlm.nih.gov/geo/query/acc.cgi?acc = GSE125373; total RNA-seq data from the same samples used for proteome analysis), GSE150149 http://www.ncbi.nlm.nih.gov/geo/query/acc.cgi?acc = GSE150149; (small RNA-seq data from the same samples used for proteome analysis), GSE124638 (http://www.ncbi.nlm.nih.gov/geo/query/acc.cgi?acc = GSE124638; ribosome foot printing data), GSE66712 (http://www.ncbi.nlm.nih.gov/geo/query/acc.cgi?acc = GSE66712; longitudinal data from (Baumgart *et al*, 2016)), and GSE150318 (http://www.ncbi.nlm.nih.gov/geo/query/acc.cgi?acc = GSE150318; longitudinal data from this study).

**Expanded View** for this article is available online.

## Acknowledgements

The authors gratefully acknowledge support from the FLI Core Facilities Proteomics, Sequencing, Imaging and Life Science Computing, and the Fish Facility. The authors would like to acknowledge Stephan Schacke for assistance in data visualization, Toby Mathieson and Holger Dinkel for assistance in establishing data analysis pipelines, Matthias Platzer for support of longitudinal RNA-seq experiment, Sabine Matz and Bernhard Schlott for assistance with experiments, and Martin Beck, K. Lenhard Rudolph, David M. Sabatini, Maria Ermolaeva, Monther Abu-Ramaileh and Aliaksandr Khaminets for critical comments on the manuscript and helpful discussion. The FLI is a member of the Leibniz Association and is financially supported by the Federal Government of Germany and the State of Thuringia. AO acknowledges funding from the Else Kröner Fresenius Stiftung (award number: 2019_A79) and the Deutsches Zentrum für Herz-Kreislaufforschung (award number: 81X2800193). AO and SDS acknowledge funding from the German Research Foundation (Deutsche Forschungsgemeinschaft, DFG) via the Research Training Group ProMoAge (GRK 2155). Research from NC laboratory is currently co-financed by the European Union and Greek national funds through the Operational Program Competitiveness, Entrepreneurship and Innovation under the call RESEARCH-CREATE-INNOVATE (project codes: T1EDK-00353 and T1EDK-01610) and under the Action "Action for the Strategic Development on the Research and Technological Sector" (project STHENOS-b, MIS 5002398). NP receives a PhD fellowship from Empirikion Foundation.

## Author contributions

Conceptualization: EKS, JMK, MM, MB, NC, MV, AC, AO; Data curation: EKS, JMK, MM, AO; Formal analysis: EKS, JMK, MM, MB, DC, MS, NR, MB, DDF, PHS, NC, AO; Investigation: EKS, JMK, MB, AB, SDS, CC, NP, ML, SB, ETT, NC; Methodology: EKS, JMK, MM, DC, WH, MV; Project administration: AC, AO; Resources: MB; Data analysis: MM, MS, DC, NR, PHS, AO; Supervision: JMK, WH, NC, MV, AC, AO; Visualization: EKS, JMK, MM, AB, SDS, MS, DC, SB, ETT, DDF, AO; Writing—original draft: EKS, JMK, MM, MV, AC, AO; Writing—review and editing: EKS, MS, ETT, NR, NC.

## Conflict of interest

The authors declare that they have no conflict of interest.

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

                                                                 ©2020 The Authors

Chondrogianni N, Petropoulos I, Franceschi C, Friguet B, Gonos ES (2000) Fibroblast cultures from healthy centenarians have an active proteasome. *Exp Gerontol* 35: 721–728

Chondrogianni N, Sakellari M, Lefaki M, Papaevgeniou N, Gonos ES (2014) Proteasome activation delays aging *in vitro* and *in vivo*. *Free Radic Biol Med* 71: 303–320

Chondrogianni N, Georgila K, Kourtis N, Tavernarakis N, Gonos ES (2015) 20S proteasome activation promotes life span extension and resistance to proteotoxicity in *Caenorhabditis elegans*. *FASEB J* 29: 611–622

Clarke LE, Liddelow SA, Chakraborty C, Munch AE, Heiman M, Barres BA (2018) Normal aging induces A1-like astrocyte reactivity. *Proc Natl Acad Sci USA* 115: E1896–E1905

Colantuoni C, Lipska BK, Ye T, Hyde TM, Tao R, Leek JT, Colantuoni EA, Elkahloun AG, Herman MM, Weinberger DR *et al* (2011) Temporal dynamics and genetic control of transcription in the human prefrontal cortex. *Nature* 478: 519–523

David DC, Ollikainen N, Trinidad JC, Cary MP, Burlingame AL, Kenyon C (2010) Widespread protein aggregation as an inherent part of aging in *C. elegans*. *PLoS Biol* 8: e1000450

Di Cicco E, Tozzini ET, Rossi G, Cellerino A (2011) The short-lived annual fish *Nothobranchius furzeri* shows a typical teleost aging process reinforced by high incidence of age-dependent neoplasias. *Exp Gerontol* 46: 249–256

Dickstein DL, Weaver CM, Luebke JI, Hof PR (2013) Dendritic spine changes associated with normal aging. *Neuroscience* 251: 21–32

Dillin A, Hsu AL, Arantes-Oliveira N, Lehrer-Graiwer J, Hsin H, Fraser AG, Kamath RS, Ahringer J, Kenyon C (2002) Rates of behavior and aging specified by mitochondrial function during development. *Science* 298: 2398–2401

Dobin A, Davis CA, Schlesinger F, Drenkow J, Zaleski C, Jha S, Batut P, Chaisson M, Gingeras TR (2013) STAR: ultrafast universal RNA-seq aligner. *Bioinformatics* 29: 15–21

Frahm C, Srivastava A, Schmidt S, Mueller J, Groth M, Guenther M, Ji Y, Priebe S, Platzer M, Witte OW (2017) Transcriptional profiling reveals protective mechanisms in brains of long-lived mice. *Neurobiol Aging* 52: 23–31

Glees P, Hasan M (1976) Lipofuscin in neuronal aging and diseases. *Norm Pathol Anat (Stuttg)* 32: 1–68

Grune T, Jung T, Merker K, Davies KJ (2004) Decreased proteolysis caused by protein aggregates, inclusion bodies, plaques, lipofuscin, ceroid, and 'aggresomes' during oxidative stress, aging, and disease. *Int J Biochem Cell Biol* 36: 2519–2530

Harel I, Benayoun BA, Machado B, Singh PP, Hu CK, Pech MF, Valenzano DR, Zhang E, Sharp SC, Artandi SE *et al* (2015) A platform for rapid exploration of aging and diseases in a naturally short-lived vertebrate. *Cell* 160: 1013–1026

Heintz C, Doktor TK, Lanjuin A, Escoubas CC, Zhang Y, Weir HJ, Dutta S, Silva-Garcia CG, Bruun GH, Morantte I *et al* (2017) Splicing factor 1 modulates dietary restriction and TORC1 pathway longevity in *C. elegans*. *Nature* 541: 102–106

Heinze I, Bens M, Calzia E, Holtze S, Dakhovnik O, Sahm A, Kirkpatrick JM, Szafranski K, Romanov N, Sama SN *et al* (2018) Species comparison of liver proteomes reveals links to naked mole-rat longevity and human aging. *BMC Biol* 16: 82

Hu CK, Brunet A (2018) The African turquoise killifish: a research organism to study vertebrate aging and diapause. *Aging Cell* 17: e12757

Janssens GE, Meinema AC, Gonzalez J, Wolters JC, Schmidt A, Guryev V, Bischoff R, Wit EC, Veenhoff LM, Heinemann M (2015) Protein biogenesis machinery is a driver of replicative aging in yeast. *eLife* 4: e08527

Khatter H, Myasnikov AG, Natchiar SK, Klaholz BP (2015) Structure of the human 80S ribosome. *Nature* 520: 640–645

Kim D, Pertea G, Trapnell C, Pimentel H, Kelley R, Salzberg SL (2013) TopHat2: accurate alignment of transcriptomes in the presence of insertions, deletions and gene fusions. *Genome Biol* 14: R36

Kim Y, Nam HG, Valenzano DR (2016) The short-lived African turquoise killifish: an emerging experimental model for ageing. *Dis Model Mech* 9: 115–129

Klus P, Bolognesi B, Agostini F, Marchese D, Zanzoni A, Tartaglia GG (2014) The cleverSuite approach for protein characterization: predictions of structural properties, solubility, chaperone requirements and RNA-binding abilities. *Bioinformatics* 30: 1601–1608

Langmead B, Trapnell C, Pop M, Salzberg SL (2009) Ultrafast and memory-efficient alignment of short DNA sequences to the human genome. *Genome Biol* 10: R25

Lee SS, Lee RY, Fraser AG, Kamath RS, Ahringer J, Ruvkun G (2003) A systematic RNAi screen identifies a critical role for mitochondria in *C. elegans* longevity. *Nat Genet* 33: 40–48

Li B, Dewey CN (2011) RSEM: accurate transcript quantification from RNA-Seq data with or without a reference genome. *BMC Bioinformatics* 12: 323

Loerch PM, Lu T, Dakin KA, Vann JM, Isaacs A, Geula C, Wang J, Pan Y, Gabuzda DH, Li C *et al* (2008) Evolution of the aging brain transcriptome and synaptic regulation. *PLoS ONE* 3: e3329

Love MI, Huber W, Anders S (2014) Moderated estimation of fold change and dispersion for RNA-seq data with DESeq2. *Genome Biol* 15: 550

Lu T, Pan Y, Kao SY, Li C, Kohane I, Chan J, Yankner BA (2004) Gene regulation and DNA damage in the ageing human brain. *Nature* 429: 883–891

Luo W, Friedman MS, Shedden K, Hankenson KD, Woolf PJ (2009) GAGE: generally applicable gene set enrichment for pathway analysis. *BMC Bioinformatics* 10: 161

Matsui H, Kenmochi N, Namikawa K (2019) Age- and α-synuclein-dependent degeneration of dopamine and noradrenaline neurons in the annual killifish *Nothobranchius furzeri*. *Cell Rep* 26: 1727–1733.e1726

Mazin P, Xiong J, Liu X, Yan Z, Zhang X, Li M, He L, Somel M, Yuan Y, Phoebe Chen YP *et al* (2013) Widespread splicing changes in human brain development and aging. *Mol Syst Biol* 9: 633

McShane E, Sin C, Zauber H, Wells JN, Donnelly N, Wang X, Hou J, Chen W, Storchova Z, Marsh JA *et al* (2016) Kinetic analysis of protein stability reveals age-dependent degradation. *Cell* 167: 803–815.e821

Morgan M, Anders S, Lawrence M, Aboyoun P, Pages H, Gentleman R (2009) ShortRead: a bioconductor package for input, quality assessment and exploration of high-throughput sequence data. *Bioinformatics* 25: 2607–2608

Myeku N, Metcalfe MJ, Huang Q, Figueiredo-Pereira M (2011) Assessment of proteasome impairment and accumulation/aggregation of ubiquitinated proteins in neuronal cultures. *Methods Mol Biol* 793: 273–296

Ori A, Banterle N, Iskar M, Andrés-Pons A, Escher C, Khanh Bui H, Sparks L, Solis-Mezarino V, Rinner O, Bork P *et al* (2013) Cell type-specific nuclear pores: a case in point for context-dependent stoichiometry of molecular machines. *Mol Syst Biol* 9: 648

Ori A, Toyama BH, Harris MS, Bock T, Iskar M, Bork P, Ingolia NT, Hetzer MW, Beck M (2015) Integrated transcriptome and proteome analyses reveal organ-specific proteome deterioration in old rats. *Cell Syst* 1: 224–237

Ori A, Iskar M, Buczak K, Kastritis P, Parca L, Andrés-Pons A, Singer S, Bork P, Beck M (2016) Spatiotemporal variation of mammalian protein complex stoichiometries. *Genome Biol* 17: 47

Platzer M, Englert C (2016) *Nothobranchius furzeri*: a model for aging research and more. *Trends Genet* 32: 543–552

Reichwald K, Petzold A, Koch P, Downie BR, Hartmann N, Pietsch S, Baumgart M, Chalopin D, Felder M, Bens M *et al* (2015) Insights into sex chromosome evolution and aging from the genome of a short-lived fish. *Cell* 163: 1527–1538

Reis-Rodrigues P, Czerwieniec G, Peters TW, Evani US, Alavez S, Gaman EA, Vantipalli M, Mooney SD, Gibson BW, Lithgow GJ *et al* (2012) Proteomic analysis of age-dependent changes in protein solubility identifies genes that modulate lifespan. *Aging Cell* 11: 120–127

Ripa R, Dolfi L, Terrigno M, Pandolfini L, Savino A, Arcucci V, Groth M, Terzibasi Tozzini E, Baumgart M, Cellerino A (2017) MicroRNA miR-29 controls a compensatory response to limit neuronal iron accumulation during adult life and aging. *BMC Biol* 15: 9

Ritchie ME, Phipson B, Wu D, Hu Y, Law CW, Shi W, Smyth GK (2015) limma powers differential expression analyses for RNA-sequencing and microarray studies. *Nucleic Acids Res* 43: e47

Schimanski LA, Barnes CA (2010) Neural protein synthesis during aging: effects on plasticity and memory. *Front Aging Neurosci* 2: 26

Schmitz C, Hof PR (2007) Design-based stereology in brain aging research. In *Brain Aging: Models, Methods, and Mechanisms*, Riddle DR (ed). Boca Raton, FL: CRC Press/Taylor & Francis Chapter 4. Available from: https://www.ncbi.nlm.nih.gov/books/NBK3880/

Schneider CA, Rasband WS, Eliceiri KW (2012) NIH Image to ImageJ: 25 years of image analysis. *Nat Methods* 9: 671–675

Schwanhäusser B, Busse D, Li N, Dittmar G, Schuchhardt J, Wolf J, Chen W, Selbach M (2011) Global quantification of mammalian gene expression control. *Nature* 473: 337–342

Sharma K, Schmitt S, Bergner CG, Tyanova S, Kannaiyan N, Manrique-Hoyos N, Kongi K, Cantuti L, Hanisch UK, Philips MA *et al* (2015) Cell type- and brain region-resolved mouse brain proteome. *Nat Neurosci* 18: 1819–1831

Shen D, Coleman J, Chan E, Nicholson TP, Dai L, Sheppard PW, Patton WF (2011) Novel cell- and tissue-based assays for detecting misfolded and aggregated protein accumulation within aggresomes and inclusion bodies. *Cell Biochem Biophys* 60: 173–185

Shim SM, Lee WJ, Kim Y, Chang JW, Song S, Jung YK (2012) Role of S5b/PSMD5 in proteasome inhibition caused by TNF-alpha/NFkappaB in higher eukaryotes. *Cell Rep* 2: 603–615

Somel M, Guo S, Fu N, Yan Z, Hu HY, Xu Y, Yuan Y, Ning Z, Hu Y, Menzel C *et al* (2010) MicroRNA, mRNA, and protein expression link development and aging in human and macaque brain. *Genome Res* 20: 1207–1218

Sormanni P, Camilloni C, Fariselli P, Vendruscolo M (2015) The s2D method: simultaneous sequence-based prediction of the statistical populations of ordered and disordered regions in proteins. *J Mol Biol* 427: 982–996

Sorrells SF, Paredes MF, Cebrian-Silla A, Sandoval K, Qi D, Kelley KW, James D, Mayer S, Chang J, Auguste KI *et al* (2018) Human hippocampal neurogenesis drops sharply in children to undetecDataset levels in adults. *Nature* 555: 377–381

Soto C, Pritzkow S (2018) Protein misfolding, aggregation, and conformational strains in neurodegenerative diseases. *Nat Neurosci* 21: 1332–1340

Stefanatos R, Sanz A (2018) The role of mitochondrial ROS in the aging brain. *FEBS Lett* 592: 743–758

Steffen KK, Dillin A (2016) A ribosomal perspective on proteostasis and aging. *Cell Metab* 23: 1004–1012

Sudmant PH, Lee H, Dominguez D, Heiman M, Burge CB (2018) Widespread accumulation of ribosome-associated isolated 3′ UTRs in neuronal cell populations of the aging brain. *Cell Rep* 25: 2447–2456.e2444

Sung MK, Porras-Yakushi TR, Reitsma JM, Huber FM, Sweredoski MJ, Hoelz A, Hess S, Deshaies RJ (2016) A conserved quality-control pathway that mediates degradation of unassembled ribosomal proteins. *eLife* 5: e19105

Terzibasi E, Valenzano DR, Benedetti M, Roncaglia P, Cattaneo A, Domenici L, Cellerino A (2008) Large differences in aging phenotype between strains of the short-lived annual fish *Nothobranchius furzeri*. *PLoS ONE* 3: e3866

Tozzini ET, Baumgart M, Battistoni G, Cellerino A (2012) Adult neurogenesis in the short-lived teleost *Nothobranchius furzeri*: localization of neurogenic niches, molecular characterization and effects of aging. *Aging Cell* 11: 241–251

Toyama BH, Savas JN, Park SK, Harris MS, Ingolia NT, Yates JR, Hetzer MW (2013) Identification of long-lived proteins reveals exceptional stability of essential cellular structures. *Cell* 154: 971–982

Valdesalici S, Cellerino A (2003) Extremely short lifespan in the annual fish *Nothobranchius furzeri*. *Proc R Soc Lond B Biol Sci* 270(Suppl 2): S189–S191

Valenzano DR, Terzibasi E, Cattaneo A, Domenici L, Cellerino A (2006a) Temperature affects longevity and age-related locomotor and cognitive decay in the short-lived fish *Nothobranchius furzeri*. *Aging Cell* 5: 275–278

Valenzano DR, Terzibasi E, Genade T, Cattaneo A, Domenici L, Cellerino A (2006b) Resveratrol prolongs lifespan and retards the onset of age-related markers in a short-lived vertebrate. *Curr Biol* 16: 296–300

Viidik A (1979) Connective tissues–possible implications of the temporal changes for the aging process. *Mech Ageing Dev* 9: 267–285

Vilchez D, Morantte I, Liu Z, Douglas PM, Merkwirth C, Rodrigues AP, Manning G, Dillin A (2012) RPN-6 determines *C. elegans* longevity under proteotoxic stress conditions. *Nature* 489: 263–268

Vizcaino JA, Csordas A, del-Toro N, Dianes JA, Griss J, Lavidas I, Mayer G, Perez-Riverol Y, Reisinger F, Ternent T *et al* (2016) 2016 update of the PRIDE database and its related tools. *Nucleic Acids Res* 44: D447–D456

Wallings RL, Humble SW, Ward ME, Wade-Martins R (2019) Lysosomal dysfunction at the centre of Parkinson's disease and frontotemporal dementia/amyotrophic lateral sclerosis. *Trends Neurosci* 42: 899–912

Walther DM, Kasturi P, Zheng M, Pinkert S, Vecchi G, Ciryam P, Morimoto RI, Dobson CM, Vendruscolo M, Mann M *et al* (2015) Widespread proteome remodeling and aggregation in aging *C. elegans*. *Cell* 161: 919–932

Wei YN, Hu HY, Xie GC, Fu N, Ning ZB, Zeng R, Khaitovich P (2015) Transcript and protein expression decoupling reveals RNA binding proteins and miRNAs as potential modulators of human aging. *Genome Biol* 16: 41

Wood SH, Craig T, Li Y, Merry B, de Magalhaes JP (2013) Whole transcriptome sequencing of the aging rat brain reveals dynamic RNA changes in the dark matter of the genome. *Age* 35: 763–776

Yanagitani K, Juszkiewicz S, Hegde RS (2017) UBE2O is a quality control factor for orphans of multiprotein complexes. *Science* 357: 472–475

Zeier Z, Madorsky I, Xu Y, Ogle WO, Notterpek L, Foster TC (2011) Gene expression in the hippocampus: regionally specific effects of aging and caloric restriction. *Mech Ageing Dev* 132: 8–19

Zoncu R, Bar-Peled L, Efeyan A, Wang S, Sancak Y, Sabatini DM (2011) mTORC1 senses lysosomal amino acids through an inside-out mechanism that requires the vacuolar H(+)-ATPase. *Science* 334: 678–683

