## [Review Process File · Molecular Systems Biology]

Reduced proteasome activity in the aging brain results in ribosome stoichiometry loss and aggregation

Alessandro Ori, Erika Kelmer Sacramento, Joanna Kirkpatrick, Mariateresa Mazzetto, Mario Baumgart, Aleksandar Bartolome, Simone Di Sanzo, Cinzia Caterino, Michele Sanguanini, Nikoletta Papaevgeniou, Maria Lefaki, Dorothee Childs, Sara Bagnoli, Eva Terzibasi Tozzini, Domenico Di Fraia, Natalie Romanov, Peter Sudmant, Wolfgang Huber, Niki Chondrogianni, Michele Vendruscolo, and Alessandro Cellerino

DOI: 10.15252/msb.20209596

Corresponding author(s): Alessandro Ori (alessandro.ori@leibniz-fli.de) , Alessandro Cellerino (alessandro.cellerino@sns.it)

Review Timeline:

Submission Date:	25th Mar 20
Editorial Decision:	27th Mar 20
Revision Received:	15th Apr 20
Editorial Decision:	23rd Apr 20
Revision Received:	12th May 20
Accepted:	13th May 20

Editor: Maria Polychronidou

Transaction Report:

The reviewers' comments and authors' responses are not available with this article, as the initial review process took place with another journal.

Thank you again for submitting your work to Molecular Systems Biology. The reviews from the other journal were quite constructive so as we discussed previously, we decided to use these reports rather than reviewing the study from scratch. Thank you for the detailed and really well-structured point by point response, it made it easy for me to evaluate the changes. I feel that the responses to the reviewers' comments seem to satisfactorily address the points raised. As most of the reviewers' concerns referred to the need to provide further support for the main conclusions, but there were no serious concerns on the technical aspects of the core experiments reported in the study, we have decided to proceed with making a decision based on our evaluation of the study and your responses.

Overall, we think that the in vivo proteasome inhibitor experiments provide additional support for the proposed role of the proteasome. Similarly the MS analysis of aggregates from young vs old mouse brains and the immunostaining in fish brain samples provide further support for the related conclusions. Several other comments referred to the need to include clarifications, text edits, methodological details and statistical support and we think that they have been satisfactorily addressed.

In summary, we think that the explanations and additional analyses performed in response to the reviewers' comments are satisfactory. As such, we have decided to proceed with the publication of your study in Molecular Systems Biology, pending some minor revisions listed below:

The Authors have made the requested editorial changes.

Thank you for sending us your revised manuscript. Before we can accept the manuscript for publication I would ask you to fix a few remaining minor issues listed below:

The Authors have made the requested editorial changes.

Thank you again for sending us your revised manuscript. We are now satisfied with the modifications made and I am pleased to inform you that your paper has been accepted for publication.

Corresponding Author Name: Alessandro Ori

Journal Submitted to: MSB

Manuscript Number: MSB-20-9596